# E2F/Dp inactivation in fat body cells triggers systemic metabolic changes

Maria Paula Zappia[1], Ana Guarner[2], Nadia Kellie-Smith[1], Alice Rogers[1], Robert Morris[2], Brandon Nicolay[2], Myriam Boukhali[2], Wilhelm Haas[2], Nicholas J Dyson[2], Maxim V Frolov[1]*

[1]University of Illinois at Chicago, Chicago, United States; [2]Massachusetts General Hospital Cancer Center and Harvard Medical School, Charlestown, United States

**Abstract** The E2F transcription factors play a critical role in controlling cell fate. In *Drosophila*, the inactivation of E2F in either muscle or fat body results in lethality, suggesting an essential function for E2F in these tissues. However, the cellular and organismal consequences of inactivating E2F in these tissues are not fully understood. Here, we show that the E2F loss exerts both tissue-intrinsic and systemic effects. The proteomic profiling of E2F-deficient muscle and fat body revealed that E2F regulates carbohydrate metabolism, a conclusion further supported by metabolomic profiling. Intriguingly, animals with E2F-deficient fat body had a lower level of circulating trehalose and reduced storage of fat. Strikingly, a sugar supplement was sufficient to restore both trehalose and fat levels, and subsequently rescued animal lethality. Collectively, our data highlight the unexpected complexity of *E2F* mutant phenotype, which is a result of combining both tissue-specific and systemic changes that contribute to animal development.

## Introduction

The mechanisms by which E2F transcription factors regulate cell cycle progression have been studied in detail. The activity of E2F1 is restrained by the retinoblastoma protein pRB, a tumor suppressor that is either mutated or functionally inactivated in various cancers (*Dyson, 2016*). In the textbook description of E2F regulation, the cyclin-dependent kinases phosphorylate pRB, releasing E2F1 to activate the expression of genes that regulate DNA synthesis, S-phase entry, and mitosis.

This textbook model, however, is incomplete. Several studies have shown that E2Fs have additional roles that extend beyond the cell cycle. Notably, E2F can also act as a regulator of cellular metabolism (*Denechaud et al., 2017*; *Nicolay and Dyson, 2013*). E2F1 was implicated in global glucose homeostasis by controlling insulin secretion in the pancreatic beta cells (*Annicotte et al., 2009*) and it is needed to promote adipogenesis (*Fajas et al., 2002*). E2F1 and pRB have been shown to form repressor complexes on the promoters of genes involved in oxidative metabolism and mitochondrial biogenesis (*Blanchet et al., 2011*), and E2F1 has been found to activate the expression of glycolytic and lipogenic genes in the liver (*Denechaud et al., 2016*). Accordingly, the inactivation of the RB pathway results in profound metabolic alterations, including changes in central carbon metabolism that confer sensitivity to oxidative stress (*Nicolay et al., 2013*; *Reynolds et al., 2014*).

The model organism *Drosophila* has provided important insights into our current understanding of E2F control. The mechanisms of action of pRB and E2F orthologues are highly conserved from flies to mammals (*van den Heuvel and Dyson, 2008*). The *Drosophila* E2F family contains two E2F genes, *E2f1* and *E2f2*. Each E2F forms a heterodimer with the binding partner Dp, which is required for high-affinity DNA binding. An important feature of the *Drosophila* E2F/RB network is that Dp is encoded by a single gene. As a result, the entire program of E2F regulation can be abolished by the inactivation of Dp, either through a *Dp* mutation or in a tissue-specific manner by RNA interference (RNAi) (*Frolov et al., 2005*; *Guarner et al., 2017*; *Royzman et al., 1997*; *Zappia and Frolov,*

*For correspondence:
mfrolov@uic.edu

Competing interests: The authors declare that no competing interests exist.

*2016*). Studies of *Dp* loss of function provide a glimpse of the overall function of E2F, a perspective that has not yet been possible in studies in mammalian cells that have far larger families of E2F, DP, and RB proteins.

The loss of E2F function, as seen in *Dp* mutants, is permissive for development until pupation when lethality occurs. This confirmed the general expectation that E2F/DP will be absolutely essential for animal viability. Surprisingly, hallmarks of E2F regulation, such as cell proliferation, differentiation, or apoptosis, were largely unaffected in *Dp* mutants (*Frolov et al., 2001*; *Royzman et al., 1997*). Interestingly, the inactivation of E2F in either the fat body, which serves the roles of liver and adipose tissue, or muscles phenocopied the lethal phenotype of *Dp* mutants pointing to the requirement of E2F for animal viability in both tissues (*Guarner et al., 2017*; *Zappia and Frolov, 2016*).

The lethality of *Dp* mutants can be rescued by tissue-specific expression of *Dp* in either the muscles or fat body (*Guarner et al., 2017*; *Zappia and Frolov, 2016*). These findings confirm the importance of Dp in these tissues but, taken together, the results are puzzling: it is not known why *Dp* expression in one of these tissues can rescue an essential function provided in the other. Answering this question is complicated by the fact that the cellular consequences of *Dp* loss in the muscles or fat body are not understood in detail. Given the inter-communication between muscle and fat body (*Demontis et al., 2014*; *Demontis and Perrimon, 2009*; *Zhao and Karpac, 2017*), it was possible that E2F inactivation in muscle might affect the fat body and/or vice versa. A further possibility was that defects resulting from tissue-specific *Dp* depletion might generate systemic changes.

To address these questions, and to ask whether the roles of E2F in muscle and fat body are the same or different, we examined the changes that occur when *Dp* is specifically removed from each of these tissues. Quantitative proteomics and metabolomic profiling of both *Dp*-deficient tissues revealed changes in carbohydrate metabolism. *Dp* deficiency in the fat body resulted in low levels of trehalose, the main circulating sugar in hemolymph, and abnormal triglycerides storage. Strikingly, these defects were suppressed on high sugar diet that also rescued the lethality of *Dp*-deficient fat body animals. Despite finding that *Dp*-deficient fat bodies and muscles share similar proteomic changes in carbohydrate metabolism, rearing larvae on high sugar diet had no beneficial effect on animals that lacked Dp function in muscles. Taken together, these findings show that E2F has important metabolic functions in both muscles and fat body and that the loss of this regulation, at least in the fat body, leads to systemic changes that can be suppressed by a high sugar diet. These observations show that E2F regulation is needed to prevent both tissue specific and systemic phenotypes.

## Results

### The loss of E2F/DP in the fat body does not impact muscle development

The expression of *UAS-Dp* RNAi transgene driven by the muscle (*Mef2-GAL4*) or fat body (*cg-GAL4*) GAL4 drivers specifically depletes Dp protein in the corresponding tissue, which results in two readily observed phenotypes. First, the inactivation of Dp in muscle is accompanied by severely reduced muscle growth in both larval and thoracic muscles (*Zappia and Frolov, 2016*). Second, fat body-specific Dp depletion triggers a reduction in fat storage and an increase in DNA damage response that promotes uncontrolled DNA replication and, ultimately, leads to the occurrence of binucleated cells (*Guarner et al., 2017*). Given that the inactivation of Dp in either tissue had a similar impact on viability and caused pupal lethality (*Guarner et al., 2017*; *Zappia and Frolov, 2016*; *Figure 1—figure supplement 1*), we asked whether Dp depletion in muscle may also affect fat body development and, conversely, if Dp deficiency in fat body may cause muscle abnormalities.

We began by comparing muscle structure between *Mef2>Dp-RNAi* and *cg>Dp-RNAi* animals. The ventral longitudinal 3 (VL3) muscles in wandering third instar larva were visualized by staining the body walls with phalloidin and 4,6-diamidino-2-phenylindole (DAPI). As previously reported, the area of the VL3 muscles in *Mef2>Dp-RNAi* animals were significantly smaller than control (*Zappia and Frolov, 2016*). In contrast, no differences were detected in the VL3 muscle area between *cg>RFP* and *cg>Dp-RNAi* larval muscles (*Figure 1A–B*). To confirm efficiency of Dp depletion in *Mef2>Dp-RNAi*, the expression of Dp was monitored with the transgene $Dp^{GFP}$, which expresses the fusion protein Dp::GFP (*Figure 1A*; *Zappia and Frolov, 2016*). Thus, we concluded that the loss of E2F/DP in fat body does not cause the phenotype observed in Dp-depleted muscles.

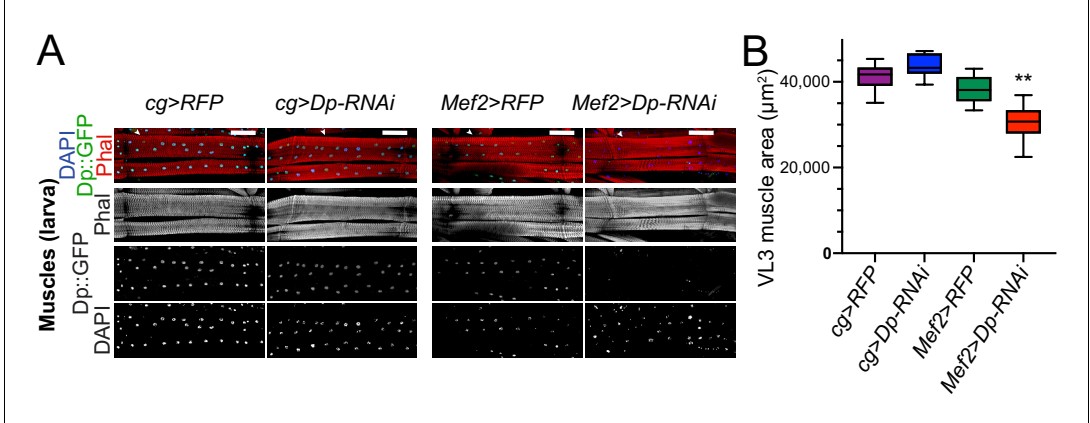

**Figure 1.** The loss of E2F in fat body does not impair muscle formation. (A) Confocal Z-stack-projected images of third instar larval body wall muscles ventral longitudinal 3 (VL3) (marked with white arrowhead) and ventral longitudinal 4 (VL4) from the segment A4 stained with Phalloidin, Dp::GFP, and DAPI. Anterior is to the left. Scale: 100 μm. (B) Box plot showing quantification of VL3 muscle area. Whiskers are min to max values, Kruskal-Wallis test followed by Dunn's multiple comparisons test, **p<0.0001. n=9–12 animals per genotype. Three independent experiments were done. Full genotypes are (A–B) *cg-GAL4/UAS-RFP,Dp[GFP]*, *cg-GAL4/Dp[GFP],UAS-Dp[GD4444]-RNAi*, *UAS-RFP,Dp[GFP];Mef2-GAL4*, and *Dp[GFP],UAS-Dp[GD4444]-RNAi, Mef2-GAL4*.

The online version of this article includes the following source data and figure supplement(s) for figure 1:

**Source data 1.** This table includes the area measurements of the ventral longitudinal 3 (VL3) muscles at the third instar larva and the statistical analysis.
**Figure supplement 1.** E2F in muscles and fat body is required for animal development.

## Dp-depleted muscles affect fat body development

One of the hallmarks of the loss of *Dp* in fat body is an increase in the DNA damage response that leads to the appearance of binucleated cells (*Guarner et al., 2017*). We confirmed that the depletion of Dp driven by the fat body-specific GAL4 driver *cg-GAL4* resulted in the formation of binucleated cells (~4.2% in *cg>Dp-RNAi*, *Figure 2A–B*) while none were found in fat bodies from wild-type animals. This phenotype was also observed in *E2f2; E2f1* double mutant animals (*Figure 2—figure supplement 1A*) and in *Dp* null mutants (*Guarner et al., 2017*). Given that the *cg-GAL4* driver expresses GAL4 in both fat body and hemocytes (*Pastor-Pareja and Xu, 2011*), an additional independent fat body driver, *Lpp-GAL4*, was used to drive Dp depletion in fat body. The occurrence of binucleated cells in fat body were also observed in *Lpp>DpRNAi* larva (*Figure 2—figure supplement 1B*), which died at pupa stage (*Figure 1—figure supplement 1*), thus confirming the cell-autonomous effect of Dp loss in fat body, in concordance with previous work (*Guarner et al., 2017*).

Next, we examined the fat body in *Mef2>Dp-RNAi* larva following muscle-specific Dp depletion by staining the tissue with phalloidin and DAPI. Surprisingly, ablation of Dp in muscles led to the appearance of binucleated cells (7.7% in *Mef2>Dp-RNAi*, *Figure 2A–B*). We confirmed that Dp was depleted in a tissue-specific manner, as only *cg>Dp-RNAi* fat bodies showed reduced levels of Dp protein as revealed by immunofluorescence using anti-Dp antibodies (*Figure 2A*, bottom panel).

One of the functions of E2F in the fat body is to limit the response to DNA damage. In Dp-depleted fat body, there is an increased recruitment of the DNA damage proteins of the MRN sensor complex, such as Rad50 and Mre11 (*Guarner et al., 2017*). To determine whether the defects found in the fat bodies of *Mef2>Dp-RNAi* animals were related to the activation of the DNA damage response, tissues were immunostained with anti-Rad50 (*Figure 2C*) and anti-Mre11 (*Figure 2—figure supplement 1C*) antibodies. Notably, the MRN proteins were not recruited in the fat body of *Mef2>Dp-RNAi* animals, as opposed to *cg>Dp-RNAi* fat body cells.

We conclude that the loss of Dp in muscles elicits defects in fat body that might be similar to the phenotype seen in Dp-deficient fat body albeit not accompanied by the upregulation of MRN proteins. Thus, Dp-deficient muscle seems to exert a systemic effect on normal tissues, such as fat body.

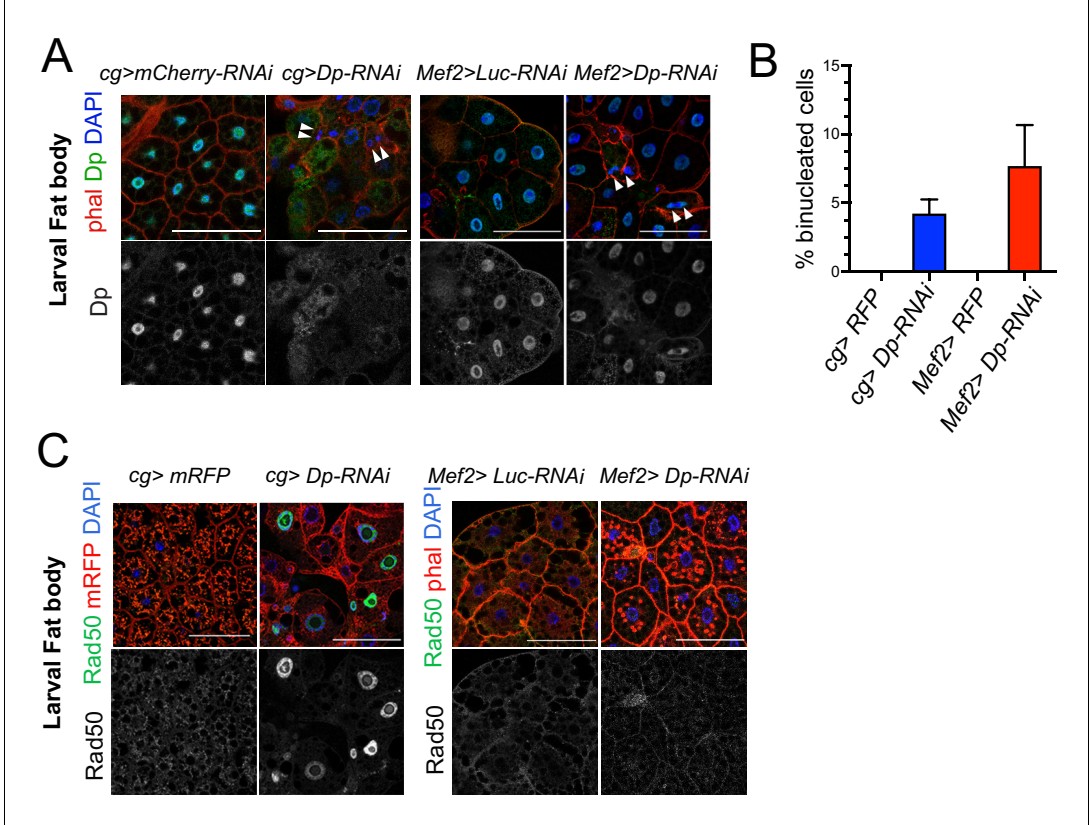

**Figure 2.** The loss of E2F in muscles has a systemic effect in the development of fat body. (**A**) Confocal single plane images of third instar larval fat bodies stained with phalloidin, 4,6-diamidino-2-phenylindole (DAPI), and mouse anti-Dp antibody (Yun). White arrowheads point to newly formed binucleate cells. (**B**) Quantification of the percentage of binucleated cells in fat body. Date are presented as mean ± SD, n=415 cells for *Mef2>Dp-RNAi*, n=405 cells for *cg>Dp-RNAi*, and n=300 cells for each *Mef2>Luc-RNAi* and *cg>Luc-RNAi*, which did not show binucleates. Kruskal-Wallis test followed by Dunn's multiple comparisons test, p<0.0001. Experiment was repeated two times. One representative experiment is shown. (**C**) Confocal single plane images of third instar larval fat bodies stained with Rad50, phalloidin, and DAPI. Scale: 100 μm. Full genotypes are (**A**) *cg-GAL4;UAS-mCherry-RNAi, cg-GAL4/UAS-Dp[GD4444]-RNAi, Mef2-GAL4/ UAS-luciferase[JF01355]-RNAi*, and *UAS-Dp[GD4444]-RNAi; Mef2-GAL4*, (**B**) *cg-GAL4/UAS-RFP,Dp [GFP], cg-GAL4/Dp[GFP],UAS-Dp[GD4444]-RNAi, UAS-RFP,Dp[GFP];Mef2-GAL4*, and *Dp[GFP],UAS-Dp[GD4444]-RNAi,Mef2-GAL4*, and (**C**) *cg-GAL4, UAS-mRFP;UAS-luciferase[JF01355]-RNAi, cg-GAL4,UAS-mRFP/UAS-Dp[GD4444]-RNAi, Mef2-GAL4/UAS-luciferase[JF01355]-RNAi*, and *UAS-Dp [GD4444]-RNAi;Mef2-GAL4*.

The online version of this article includes the following figure supplement(s) for figure 2:

**Figure supplement 1.** Loss of E2F induces binucleated cells in fat body.

## The loss of Dp in muscles does not alter Dp expression in the fat body

Leaky expression of *GAL4* drivers in other tissues during earlier developmental stages might provide a trivial explanation for the results described above. To exclude this possibility, we used three approaches to confirm the tissue specificity of *cg-GAL4* and *Mef2-GAL4* drivers used to knock down Dp. First, we examined the real-time and lineage tracing expression of the drivers. We used the system G-TRACE, which combines the FLP recombinase-FRT and the expression of GFP to trace earlier GAL4 expression, and the presence of RFP to identify real-time expression of GAL4 (*Evans et al., 2009*). G-TRACE showed that the *cg-GAL4* driver is expressed in the fat body in agreement with previous report (*Pastor-Pareja and Xu, 2011*; *Figure 3A*, left panel). No GFP or RFP signal was detected in larval muscles (*Figure 3A*, right panel) suggesting that at no point during development *cg-GAL4* was expressed in muscles. Similarly, *Mef2-GAL4* expression was detected in larval skeletal and smooth muscles (*Figure 3B*, left panels), and in the adult muscle precursors (wing disc myoblasts), but not in the fat body (*Figure 3B*, right panels).

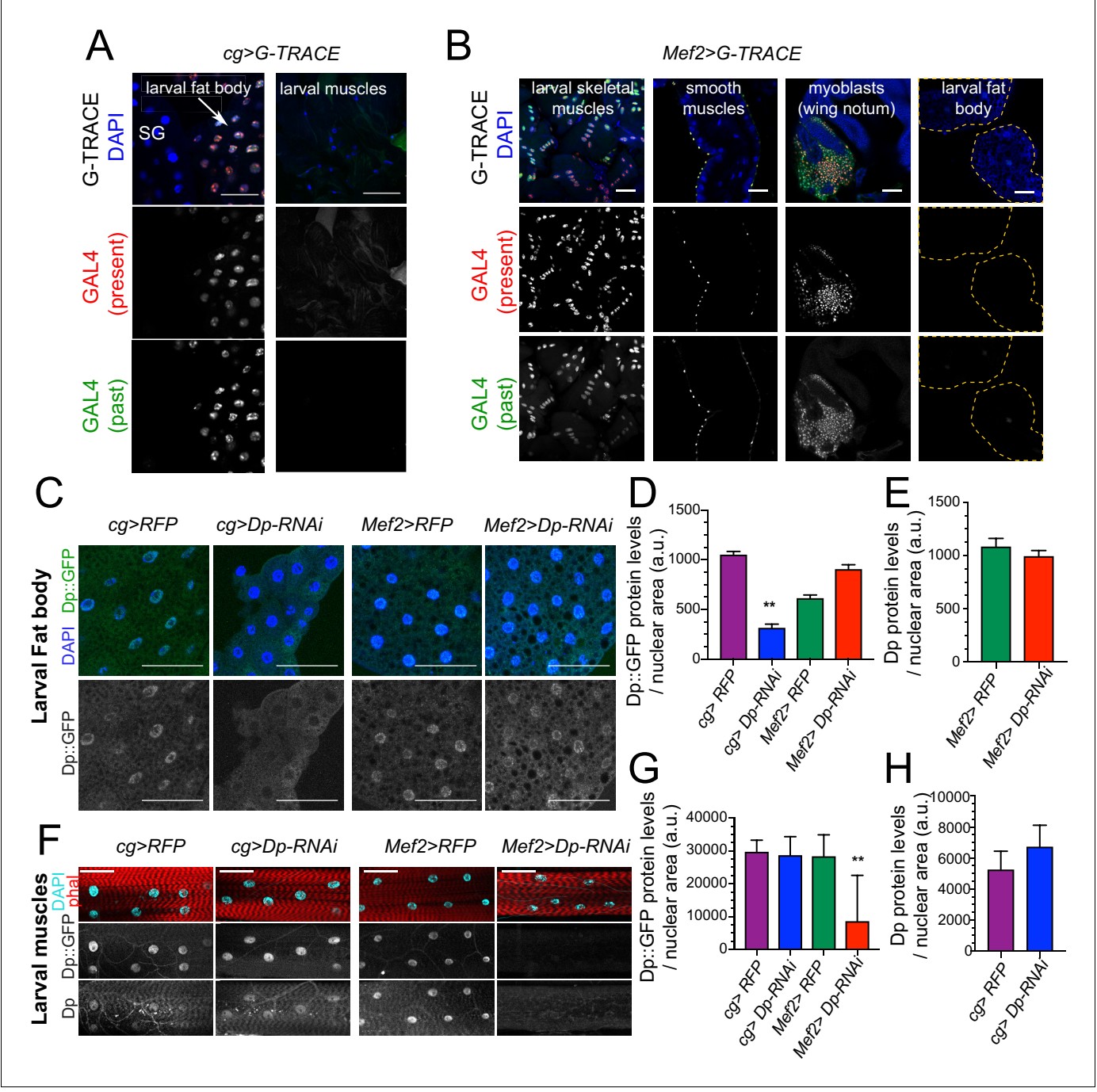

**Figure 3.** Dp expression is knock down in a tissue-specific manner. (A–B) Lineage tracing of tissues dissected from third instar larvae stained with 4,6-diamidino-2-phenylindole (DAPI) and showing the lineage of cg-GAL4 (GFP) and the active GAL4 (RFP) . (A) Confocal single plane images of *cg>G-TRACE* fat bodies, salivary glands, and muscles. Scale: 100 μm. (B) Confocal single plane images of *Mef2>G-TRACE* larval skeletal (body wall) and smooth (gut) muscles, adult myoblasts on the wing discs, and fat bodies. Scale: 50 μm. (C) Confocal single plane images of third instar larval fat bodies stained with DAPI and showing Dp::GFP tagged protein. White arrowheads indicate binucleated cells. Scale: 100 μm. (D) Quantification of Dp::GFP levels as shown in C, relative to nuclear area. Mean ± SEM, Kruskal-Wallis test followed by Dunn's multiple comparisons test, **p<0.0001, n=11–20 per genotype. (E) Quantification of Dp protein levels relative to nuclear area in larval fat body of *Mef2>RFP* and *Mef>Dp-RNAi* animals. Mean ± SEM, Mann-Whitney test, p = 0.35, n=14 per genotype. (F) Confocal single plane images of third instar larval muscles immunostained with rabbit anti-Dp antibody (212), phalloidin, and DAPI. Scale: 50 μm. (G) Quantification of Dp::GFP levels as shown in F, relative to nuclear area. Mean ± SD, Kruskal-Wallis test followed by Dunn's multiple comparisons test, **p = 0.0008, n=10–12 animals per genotype. (H) Quantification of Dp protein levels relative to nuclear area in larval fat body of *cg>RFP* and *cg>Dp-RNAi* animals. Mean ± SD, Mann-Whitney test, p = 0.07, n=6–9 per genotype. Full genotype:

*Figure 3 continued on next page*

Figure 3 continued

(A) *cg-GAL4/UAS-gTRACE*, (B) *UAS-gTRACE,Mef2-GAL4*, (C–H) *cg-GAL4/UAS-mRFP,Dp[GFP]*, *cg-GAL4/Dp[GFP],UAS-Dp [GD4444]-RNAi, UAS-mRFP, Dp[GFP];Mef2-GAL4*, and *Dp[GFP],UAS-Dp[GD4444]-RNAi,Mef2-GAL4*.

Second, the levels of Dp expression in fat body and muscles of *Mef2>Dp-RNAi* and *cg>Dp-RNAi* larva were examined by immunofluorescence using Dp antibodies. Third, Dp expression was monitored using a *Dp^GFP* transgene that expresses a Dp::GFP fusion protein from the endogenous *Dp* locus (*Zappia and Frolov, 2016*). The Dp::GFP was efficiently depleted in fat body of *cg>Dp-RNAi* compared to control (*Figure 3C*, quantified in *Figure 3D*). Importantly, the levels of the Dp::GFP remained unaffected in the fat body of *Mef2>Dp-RNAi* (*Figure 3C*, quantified in *Figure 3D*). This result was further confirmed by staining with anti-Dp antibody that showed no changes in the endogenous expression of Dp in fat body of *Mef2>Dp-RNAi* (*Figure 3E*). Similarly, the expression of Dp::GFP in muscles was not altered in *cg>Dp-RNAi*, whereas, as expected, Dp::GFP was significantly reduced in muscles of *Mef2>Dp-RNAi* compared to control (*Figure 3F*, quantified in *Figure 3G*). Using anti-Dp antibody, we further confirmed that the endogenous levels of Dp protein in muscles did not change upon Dp depletion in the fat body (*Figure 3F*, quantified in *Figure 3H*).

Thus, the occurrence of binucleated cells in *Mef2>Dp-RNAi* fat body is not due to altered expression of Dp in fat body of these animals and, therefore, reflects a systemic effect induced by muscle-specific Dp depletion.

## The muscle-specific expression of Dp in *Dp* mutants rescues the fat body phenotype

The expression of *UAS-Dp* transgene with either the fat body- or muscle-specific drivers, *cg-GAL4* or *Mef2-GAL4*, can significantly extend viability of *Dp* mutants (*Guarner et al., 2017*; *Zappia and Frolov, 2016*). Given the systemic effect of Dp described above, we asked whether the muscle-specific Dp expression suppresses the defects in fat body of *Dp* mutants. The *UAS-Dp* transgene was expressed in the trans-heterozygous *Dp^a3/Df(2R)Exel7124* (*Dp-/-*) mutant animals under the control of *cg-GAL4* or *Mef2-GALl4*. Larval fat bodies were stained with phalloidin and DAPI to visualize the occurrence of binucleated cells. In agreement with previously published data (*Guarner et al., 2017*), 7.1% of cells in fat body of *Dp* mutants were binucleated and this phenotype was fully rescued in the *Dp-/-; cg>Dp* animals (*Figure 4A*, quantified in *Figure 4B*). Strikingly, the binucleated phenotype was also largely rescued by re-expression of *Dp* in muscles of *Dp* mutants, in *Dp-/-; Mef2>Dp* animals (*Figure 4A–B*). We note, however, that fat bodies of *Dp-/-; Mef2>Dp* animals still contained fragmented and decondensed/large nuclei indicating that the rescue was incomplete. Staining with anti-Dp antibody confirmed the lack of Dp expression in *Dp-/-; Mef2>Dp* fat bodies (*Figure 4A*, bottom panel), thus excluding the possibility of a leaky expression of *Mef2-GAL4* driver in the fat body.

Next, we asked whether re-expression of *Dp* in the fat body suppresses muscle defects of *Dp* mutants. The body walls of third instar larvae were dissected and VL3 muscles of the A4 segment were visualized by staining the tissue with DAPI and phalloidin. As previously reported, VL3 muscle area was smaller in *Dp-/-* mutant larvae and fully rescued in *Dp-/-; Mef2>Dp* (*Figure 4C*, quantified in *Figure 4D*; *Zappia and Frolov, 2016*). In contrast, the expression of *Dp* in the fat body was insufficient to suppress the small size of VL3 muscles in *Dp-/-* mutant larvae (*Figure 4C*, quantified in *Figure 4D*).

Thus, muscle-specific expression of *Dp* can rescue the binucleated phenotype of *Dp* mutant fat body, which is consistent with the idea that E2F/Dp in muscle exerts a systemic effect in larva that impacts the fat body. In contrast, re-expressing *Dp* in the fat body is insufficient to suppress the muscle defects in *Dp* mutants. This suggests that E2F/Dp modifies the inter-tissue communication between muscle and fat body.

## Integrating proteomic and metabolomic profiling of E2F-depleted tissues uncovers alterations in carbohydrate metabolism

A significant limitation of transcriptional profiles is that it is difficult to know whether a change in mRNA levels leads to a measurable difference in protein level or causes a change in pathway activity.

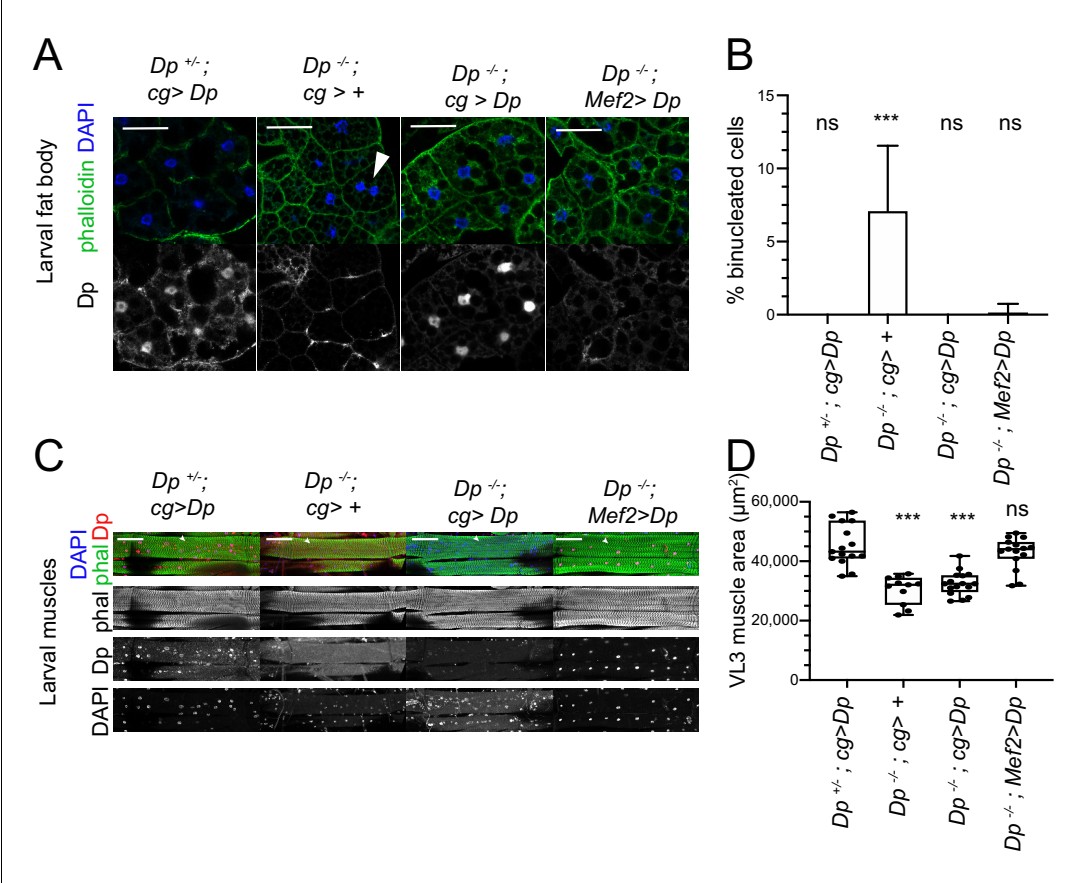

**Figure 4.** Restoring E2F/Dp in muscles suppresses defect in *Dp*-deficient fat body. (**A**) Confocal single plane images of third instar larval fat bodies immunostained with anti-Dp antibody (212), phalloidin, and 4,6-diamidino-2-phenylindole (DAPI). Note that *Dp*<sup>-/-</sup>;*Mef2>Dp* rescued animals do not show binucleated cells, but many nuclei are fragmented and decondensed. Scale: 50 µm. (**B**) Quantification of percentage of binucleated cells as in A. Data presented as bar plot showing mean ± SD, Kruskal-Wallis test followed by Dunn's multiple comparisons test, ***p<0.0001, n=16 animals per genotype, two independent experiments were done. At least 606 cells were scored. (**C**) Confocal Z-stack-projected images of third instar larval body wall muscles ventral longitudinal 3 (VL3) (marked with white arrowhead) and ventral longitudinal 4 (VL4) from the segment A4 immunostained with rabbit anti-Dp antibody (212), phalloidin, and DAPI. Anterior is to the left. Scale: 100 µm. (**D**) Quantification of VL3 muscle area as in C. Data presented as box plot, whiskers min to max values, Kruskal-Wallis test followed by Dunn's multiple comparisons test, ***p<0.0001, n=15 animals per genotype, except n=11 for Dp-/-, two independent experiments were done. Full genotypes are $Dp^+/Dp^{a3}$,cg-GAL4;UAS-Dp, $Dp^{Exel7124}/Dp^{a3}$,cg-GAL4, $Dp^{Exel7124}/Dp^{a3}$,cg-GAL4;UAS-Dp, $Dp^{Exel7124}/Dp^{a3}$; Mef2-GAL4/UAS-Dp.

The online version of this article includes the following source data for figure 4:

**Source data 1.** This table includes the area measurements of the ventral longitudinal 3 (VL3) muscles at the third instar larva and the statistical analysis.

To avoid these issues, we generated proteomic profiles of fat body and skeletal muscles and used these to compare how the loss of E2F affects protein levels in each tissue. Third instar larval fat bodies were collected from both wild type ($w^{1118}$) and *Dp-/-* mutant (*Figure 5A*, left panel) and subjected to multiplexed quantitative mass spectrometry-based proteomics using tandem-mass tag (TMT) technology (*Edwards and Haas, 2016*; *McAlister et al., 2014*). Collecting larval muscles in sufficient quantities for such proteomic profiling was not feasible due to technical challenge of separating larval muscle from adjacent tissue. Therefore, we turned to dissecting thoracic muscles from pharate pupa. We used *Mef2>Dp-RNAi* pharates since *Dp-/-* mutants die as early pupa (*Figure 5A*, right panel). Western blot analysis confirmed that the levels of Dp protein were low in lysates from *Dp-/-* fat bodies and from Dp-depleted muscles compared to controls (*Figure 5—figure supplement 1A*). Note that two distinct developmental stages are being compared in our study, and it could potentially introduce additional variations when comparing data between fat body and muscles.

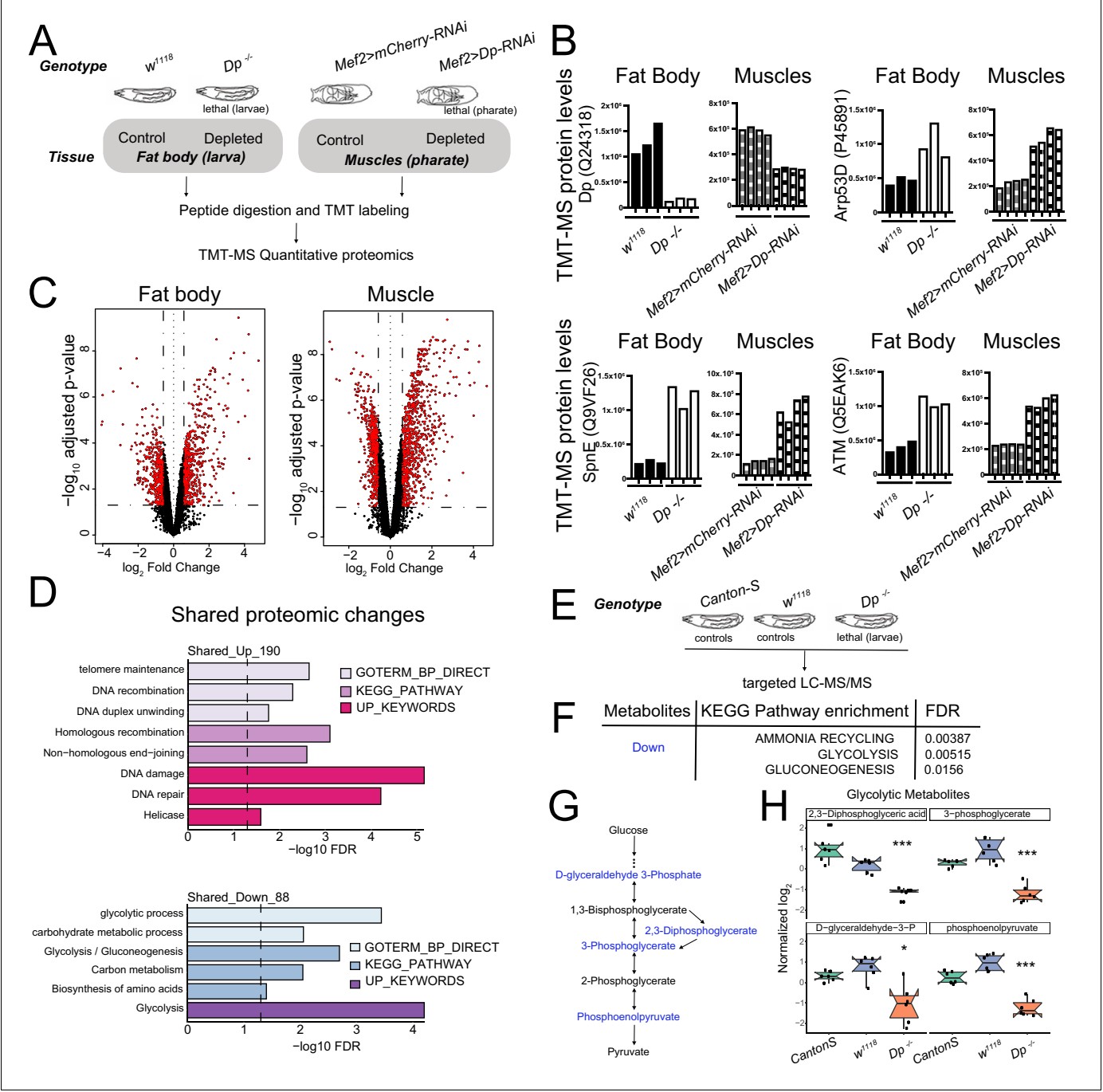

**Figure 5.** Loss of E2F/Dp impairs carbohydrate metabolism. (A) Overview of tandem-mass tag-mass spectrometry (TMT-MS) profiles generated from third instar larval fat body in *Dp-/-* mutant and wild-type (WT) animals and from thoracic muscles in *Mef2>Dp-RNAi* and control animals staged at pharate; 6578 identified proteins in fat bodies and 5730 identified proteins in muscles. (B) TMT-MS intensities showing protein levels of Dp (Uniprot Q24318), E2F2/DP target protein Arp53 (Uniprot Q9VF26), SpnE (Uniprot P455891), and dATM (Uniprot Q5EAK6) in fat bodies and muscles. Data are represented as individual intensity value for each replicate, n = 3 per genotype in fat body and n=4 per genotype in muscles. (C) Volcano plots indicating proteins that are differentially expressed between larval WT and *Dp-/-* mutant fat bodies (left panel), and *Mef2>mCherry-RNAi* and *Mef2>Dp-RNAi* pharate muscles (right panel). Significant changes are shown in red (false discovery rate [FDR] < 0.05 and abs [fold change]>1.5). The x-axis is the $\log_2$ of the fold change and the y-axis is the negative $\log_{10}$ of the adjusted p-value. (D) DAVID functional annotation clustering analysis of proteomic changes in Dp-depleted tissue compared to WT that are in common between fat body and muscles. Upregulated (top panel, 190 proteins) and downregulated (bottom panel, 88 proteins) were analyzed separately. Dashed line indicates FDR=0.05. Only significant terms (FDR<0.05) are displayed. The categories GO term biological processes, KEGG pathway, and Up keywords are shown. (E) Overview of targeted liquid chromatography tandem mass spectrometry (LC-MS/MS) metabolomic profiles generated from whole third instar larvae. *Dp-/-* mutant animals were compared to two

*Figure 5 continued on next page*

Figure 5 continued

different WT animals (Canton-S and $w^{1118}$). (F) KEGG pathway enrichment was done on metabolites that were significantly reduced in *Dp-/-* mutant compared to both controls. Only significant terms are shown (FDR<0.05). (G) Schematic of the flow of glycolysis toward pyruvate. Metabolites that are significantly reduced in *Dp-/-* mutant compared to controls are shown in blue. (H) Normalized levels of the metabolites D-glyceraldehyde 3-phosphate, 2,3-diphosphoglycerate, 3-phosphoglycerate, and phosphoenolpyruvate in *Dp-/-* mutant compared to controls $w^{1118}$ and Canton-S. Data are represented as box plot, which extends from 25 to 75 percentiles; line marks median; whiskers extend to 25/75% plus 1.5 times the interquartile range. Values outside the whiskers are outliers. Welch's ANOVA test, *p<0.05, *** p<0.001. n=6 per genotype. Full genotypes are (A–D) $Dp^{a2}/Dp^{a3}$, $w^{1118}$, *UAS-Dp[GD4444]-RNAi,Mef2-GAL4* and *Mef2-GAL4/UAS-mCherry-RNAi* (E–H) $Dp^{Exel7124}/Dp^{a3}$, $w^{1118}$ and *Canton-S.*

The online version of this article includes the following figure supplement(s) for figure 5:

**Figure supplement 1.** E2F/Dp-deficient muscles and fat bodies undergo severe changes in their proteome and metabolome.

The changes in the proteomic profiles of fat bodies between the control, $w^{1118}$, and *Dp* mutant larva, and in muscles between the control, *Mef2>mCherry-RNAi*, and *Mef2>Dp-RNAi* pharates were examined (*Supplementary file 1*). The pairwise correlation between replicate samples was found to be significant (*Figure 5—figure supplement 1B*). The mean of correlation values for fat body replicates was r=0.55 for $w^{1118}$ and r=0.42 for *Dp-/-*, while muscle replicates showed r=0.8 for *Mef2>mCherry-RNAi* and r=0.63 for *Mef2>Dp-RNAi*.

In the proteomics datasets, we identified and quantified 6578 proteins in fat body and 5370 proteins in muscle samples. We confirmed that the intensity level of Dp protein was downregulated in both *Dp-/-* fat bodies and Dp-depleted muscles compared to the control tissues, respectively. Concordantly, the expression levels of the well-known E2F-direct targets that are repressed by E2F, such as *Arp53D*, *SpnE*, and *dATM*, were upregulated in both *Dp-/-* fat bodies and Dp-depleted muscles (*Figure 5B*).

A set of 5024 proteins that were shared between both tissues was selected for integrative analysis (*Supplementary file 2*). A log$_2$fold change (FC) ratio was calculated to identify proteins that were changed upon the loss of *Dp*. These two datasets showed a Pearson correlation of r = 0.172 (*Figure 5—figure supplement 1C*) indicating that a relevant subset of the changes detected in fat body were also present in muscles. Statistically significant changes were revealed using a false discovery rate (FDR) < 5%, and either a log$_2$FC > 0.5 or a log$_2$FC < −0.5 as cutoff values for upregulated and downregulated proteins, respectively (*Figure 5C*). We found that 556 proteins increased and 456 proteins decreased in *Dp-/-* fat bodies, whereas 844 increased and 1208 proteins decreased in Dp-depleted muscles (these are visualized in heatmaps in *Figure 5—figure supplement 1D–E*).

KEGG pathways enrichment analysis was performed using the functional annotation tool DAVID (*Figure 5—figure supplement 1F*, *Supplementary file 3*) to obtain an overall picture of the changes resulting from the loss of E2F/Dp in the two tissues. Categories related to *DNA repair*, *glutathione metabolism*, and *amino acid metabolism* were significantly enriched for upregulated proteins in both tissues, while *nucleotide metabolism* was only significantly enriched in muscle (FDR<5%, *Figure 5—figure supplement 1F*, top panel). Similar changes have been linked to the loss of E2F/Dp in previous studies (*Guarner et al., 2017*; *Nicolay et al., 2015*; *Nicolay et al., 2013*). Additionally, proteins related to *cytochrome p450 enzymes* that catalyze detoxification and biosynthetic reactions in fat body (*Chung et al., 2009*) were upregulated in *Dp* mutants (*Figure 5—figure supplement 1F*, top panel). Among the downregulated proteins, the fat body proteome was enriched for *pentose phosphate pathway* and *fatty acid metabolism*, whereas *citrate cycle*, *oxidative phosphorylation*, and *ribosome* categories were significantly overrepresented among the muscle proteome upon the loss of Dp (*Figure 5—figure supplement 1F*, bottom panel).

Next, we focused on proteomic changes that were shared between these two tissues since these may reflect an E2F function that is common to both tissues (*Figure 5D*, *Supplementary file 4*). As expected, the upregulated proteins in Dp-deficient fat body and muscle showed a significant enrichment for *DNA damage*, *DNA recombination*, and *homologous recombination* (*Figure 5D*, top panel, FDR < 5%). The top annotation cluster for downregulated proteins displayed a significant enrichment for *glycolysis*, *gluconeogenesis*, and *carbohydrate metabolic process* (*Figure 5D*, bottom panel, FDR < 5%), thus indicating that the loss of Dp alters carbohydrate metabolism in both fat body and muscle.

To explore the metabolic defects triggered by the loss of Dp, we used targeted liquid chromatography tandem mass spectrometry (LC-MS/MS) to profile the metabolic changes upon *Dp* loss.

Third instar $Dp^{-/-}$ larva were collected and compared to two wild-type strains, $w^{1118}$ and *Canton S*, to account for differences in the genetic background (*Figure 5E*, *Supplementary file 5*). Fifty-five compounds showed significant changes in the *Dp* mutant compared to both controls (FDR < 5%, *Figure 5—figure supplement 1G*, *Supplementary file 6*). The increased and decreased metabolites were selected, and KEGG pathways enrichment analysis was performed. Interestingly, the major metabolic pathways that showed significant enrichment for downregulated compounds were *glycolysis*, *gluconeogenesis,* and *ammonia recycling* (FDR < 5%, *Figure 5F*, *Supplementary file 7*), which is largely consistent with the proteome analysis described above. Notably, four metabolites of the core module of the glycolytic pathway, (*Figure 5G*), 2,3-diphosphoglyceric acid, 3-phosphoglycerate, D-glyceraldehyde-3-phosphate, and phosphoenolpyruvate, were significantly reduced in *Dp* mutant compared to both controls (*Figure 5H*).

We conclude that the tissue-specific depletion of *Dp* results in extensive metabolic changes in both fat body and muscle. These changes were evident in proteomic profiles and confirmed by metabolomic profiling. The alterations indicate that E2F-depleted tissues undergo significant changes in carbohydrate metabolism affecting, in particular, glycolytic metabolites.

## Increasing carbohydrates in fly diet rescues the lethality caused by the loss of Dp in fat body

The changes in the proteomic and metabolomic profiles are very interesting but they raised the question of whether the metabolic changes observed contribute to the lethality of *Mef2>Dp-RNAi* or *cg>Dp-RNAi* animals. Since diet is known to impact metabolic phenotypes, we asked whether varying the levels of carbohydrates, protein, and fat in fly food could alter the lethal stage of these animals.

To properly control the food composition and the effect of nutrients, we switched to a semi-defined food, made of sucrose, lecithin, and yeast, as major sources of carbohydrates, fat, and protein, respectively (*Lee and Micchelli, 2013*). Control diet contained 7.9% carbohydrates, 0.08% fat, and 1.9% protein (*Supplementary file 8*). As expected, *Mef2>Dp-RNAi* or *cg>Dp-RNAi* did not survive on control diet and died at pupal and pharate stage (*Figure 6A–B*). Nutrient composition was then altered by varying the amount of a single component of the control diet and the viability of *cg>Dp-RNAi* and *Mef2>Dp-RNAi* animals relative to the control genotype was scored. Interestingly, while the survival of *Mef2>Dp-RNAi* animals was unaffected by different nutrient composition (*Figure 6B*, and *Figure 6—figure supplement 1B*), *cg>Dp-RNAi* were highly sensitive to dietary changes. The increase in protein content had a negative impact on the survival of *cg>Dp-RNAi* and resulted in significant developmental delay (*Figure 6—figure supplement 1A*, left panel). About 83% of animals showed melanotic masses at larval stages when reared on high protein diet, and only 35% of third instar larva progressed onto pupa stages and eventually died (*Figure 6—figure supplement 1C*, quantified in *Figure 6—figure supplement 1D*). The melanotic masses, which were also observed in the *E2f1* mutant larvae (*Royzman et al., 1997*), are related to the immune system response (*Watson et al., 1991*). However, reducing the content of protein suppressed the occurrence of melanotic masses (*Figure 6—figure supplement 1C–D*) and consequently increased their survival rate (*Figure 6—figure supplement 1A*). In contrast, increasing lipid content up to 1% in diet was beneficial for survival as about 22% of *cg>Dp-RNAi* pupa reached adulthood (*Figure 6—figure supplement 1A*, right panel). However, no further rescue in viability was detected beyond 1% lipid. Strikingly, while no *cg>Dp-RNAi* animals survived on control diet (7.9% carbohydrates), almost half of them eclosed when reared in the presence of 12% carbohydrates and the lethality was fully rescued when food contained even higher amount of carbohydrates (16% or 24%) (*Figure 6A*, left panel). This result was confirmed using a second fat body-specific driver *r4-GAL4* (*Figure 6A*, right panel).

One of the functions of fat body is to maintain homeostasis of trehalose, a main circulating sugar in hemolymph (*Becker et al., 1996*), allowing animals to adapt to a high sugar diet. The synthesis of trehalose occurs in fat body from glucose-6P and is regulated by trehalose-6-phosphate synthase (Tps1) (*Figure 6C*; *Elbein et al., 2003*). We noted that the levels of Tps1 proteins were significantly reduced in *Dp* mutant fat body, along with other enzymes that generate glucose-6P including phosphoglucomutase, glucose-6-phosphate isomerase, and hexokinase_A in the proteomic datasets (*Figure 6C* and *Supplementary file 2*). Additionally, trehalase, an enzyme that converts trehalose

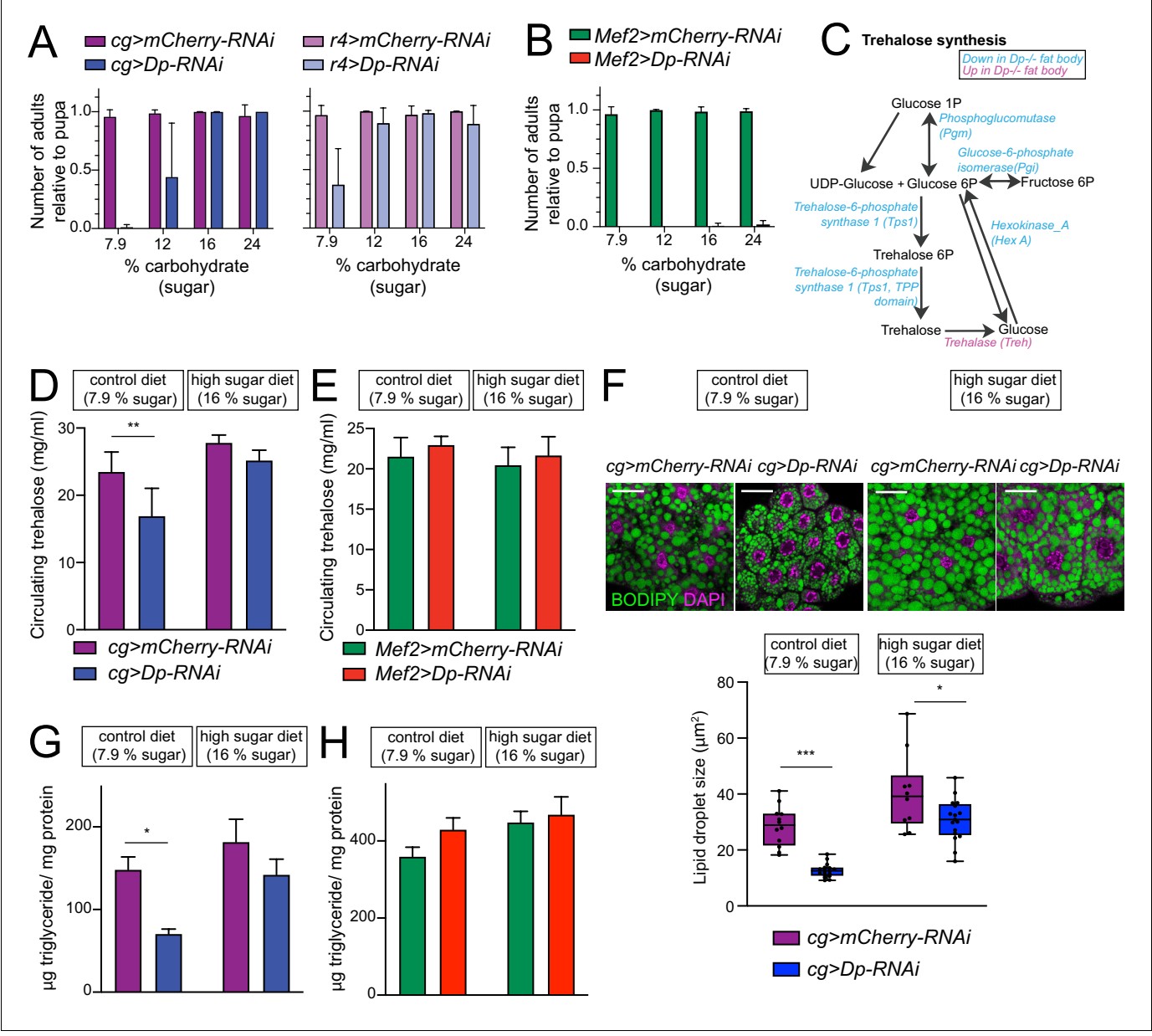

**Figure 6.** E2F/Dp in fat body exerts systemic effects modulated by sugar supplement. (**A–B**) Number of viable adults (relative to pupa) fed on control diet (7.9% carbohydrate, 0.08% fat, and 1.9% protein) and increasing levels of sugar in food (12, 16% and 24% carbohydrate). Data are represented as mean ± SD, n = 6 repeats per condition. (**A**) Left panel: *cg>mCherry RNAi* and *cg>DpRNAi*, right panel: *r4>mCherryRNAi* and *r4>DpRNAi*. (**B**) *Mef2>mCherry-RNAi* and *Mef2>Dp-RNAi*. (**C**) Diagram of trehalose synthesis pathway. The enzymes that are significantly downregulated in *Dp*-deficient fat body are indicated in blue, and upregulated in magenta, based on proteome data. (**D–E**) Circulating trehalose levels measured in third instar larval hemolymph. (**D**) *cg>mCherry-RNAi* and *cg>Dp-RNAi*, and (**E**) *Mef2>mCherry-RNAi* and *Mef2>Dp-RNAi* larvae fed on control diet (7.9% carbohydrate) and high sugar diet (16% carbohydrate). Data are represented as mean ± SD, two-way ANOVA followed by Tukey's multiple comparisons test, three independent experiments were done, one representative experiment is shown. (**D**) n = 6 per group and **p = 0.0005. (**E**) n = 3–6 per group and p = 0.5. (**F**) Top panel: Confocal single plane images of third instar larval fat bodies stained with 4,6-diamidino-2-phenylindole (DAPI) and BODIPY red. The *cg>mCherry-RNAi* and *cg>Dp-RNAi* animals were fed on control diet (7.9% carbohydrate) and supplemented with sugar (16% carbohydrate, high sugar diet). Scale: 40 μm. Bottom panel: Measurement of lipid droplet size in fat body. Data are represented as box and whiskers, min to max showing all points, n = 10–17 fat bodies per genotype, two-way ANOVA followed by Tukey's multiple comparisons test, *p = 0.02, ***p < 0.0001, three independent experiments, one representative experiment is shown. (**G–H**) Triglycerides content measured in third instar larva and normalized to total protein content. (**G**) *cg>mCherry RNAi* and *cg>DpRNAi*, and (**H**) *Mef2>mCherry RNAi* and *Mef2>DpRNAi* larvae fed on control diet (7.9% carbohydrate) and high sugar diet (16% carbohydrate). Data are represented as mean ± SEM, two-way ANOVA followed by Tukey's multiple comparisons test, one representative experiment is shown, (**G**) n = 5–6 per group and *p = 0.004. Three independent experiments were done. (**H**) n =

*Figure 6 continued on next page*

Figure 6 continued

5–6 per group and p = 0.2, two independent experiments were done. Full genotypes are (**A**) *cg-GAL4,UAS-mCherry-RNAi, cg-GAL4/UAS-Dp[GD4444]-RNAi, r4-GAL4/UAS-mCherry-RNAi, UAS-Dp[GD4444]-RNAi/r4-GAL4* (**B, E, H**) *Mef2-GAL4/UAS-mCherry-RNAi,* and *UAS-Dp[GD4444]-RNAi,Mef2-GAL4* (**D, F, G**) *cg-GAL4,UAS-mCherry-RNAi* and *cg-GAL4/UAS-Dp[GD4444]-RNAi.*

The online version of this article includes the following source data and figure supplement(s) for figure 6:

**Source data 1.** This table includes the measurements of circulating trehalose levels and the statistical analysis.
**Source data 2.** This table includes the measurements of triglycerides levels normalized to protein content and the statistical analysis.
**Source data 3.** This table includes the average values of the lipid droplet size measured for each image and the statistical analysis.
**Source data 4.** Macro analyze particle size- LipidDroplets_Script_ImageJ.
**Figure supplement 1.** Supplementing food modifies viability of animals with E2F/Dp-deficient fat bodies.

back to glucose, was significantly increased in *Dp*-deficient fat body. This suggested that animals lacking Dp in the fat body may be defective in the regulation of trehalose synthesis.

To test this idea directly, trehalose was measured in hemolymph of *cg>Dp-RNAi* and control, *cg>mCherry-RNAi*, third instar larvae fed either a control diet or a high sugar diet containing 16% sugar. Notably, trehalose was significantly reduced in hemolymph of *cg>Dp-RNAi* larva compared to matching control (*Figure 6D*) when animals were reared on control diet. Switching to high sugar diet significantly improved the levels of trehalose in *cg>Dp-RNAi* to levels comparable to *cg>mCherry-RNAi* larva that were fed a control diet (*Figure 6D*). In contrast, the levels of circulating trehalose were largely unchanged in *Mef2>Dp-RNAi* compared to control (*Figure 6E*).

The fat body is the principal site of stored fat in *Drosophila. Dp* mutants were previously shown to have lipid droplets of reduced and irregular size (*Guarner et al., 2017*). Therefore, we examined the impact of high sugar diet on fat body in *cg>Dp-RNAi*. Fat bodies of *cg>Dp-RNAi* and control larva were dissected and stained with BODIPY to visualize lipid droplets. In control diet, Dp-depleted fat bodies displayed smaller lipid droplet size compared to control (*Figure 6F*, quantification in bottom panel), which is indicative of defects in fatty acid synthesis. Concordantly, the expression of acetyl-CoA carboxylase, the rate-limiting enzyme for fatty acid synthesis, was significantly reduced in fat body of *cg>Dp-RNAi* compared to control (*Supplementary file 2*). As expected (*Musselman et al., 2011*; *Pasco and Léopold, 2012*), feeding control larva a high sugar diet resulted in significantly larger lipid storage droplets (*Figure 6F*). A similar trend was observed in *cg>Dp-RNAi* although the increase in droplet size was not as robust as in the wild type (*Figure 6F*, quantification in bottom panel). Nevertheless, feeding *cg>Dp-RNAi* animals on high sugar diet dramatically improved the lipid droplet size in comparison to control diet. This was further corroborated by measuring total triglycerides content at third instar larval development (*Figure 6G*). The levels of triglycerides normalized to total protein content were significantly reduced in *cg>Dp-RNAi* animals reared on control diet compared to *cg>mCherry-RNAi*. Moreover, the normalized levels of triglycerides were fully restored in *cg>Dp-RNAi* animals fed on high sugar diet. Thus, feeding animals with high content of sugar increased their lipid storage, as evidenced by examining both the lipid droplet size in fat body and the total content of triglycerides, and could, subsequently, contribute to rescue animal viability upon *Dp* knockdown in fat body.

In contrast, muscle-specific Dp depletion did not affect either the size of lipid droplets in the fat body or the total content of triglycerides in *Mef2>Dp-RNAi* animals (*Figure 6—figure supplement 1G* and *Figure 6H*, respectively). The *Mef2>Dp-RNAi* animals reared on high sugar showed subtle changes although not statistically different from control animals (*Figure 6H* and *Figure 6—figure supplement 1E*, quantification in bottom panel). This suggests that the loss of Dp in muscle does not severely impair the ability of fat body to regulate fat storage.

Dp-depleted fat body failed to maintain the proper level of circulating sugar in hemolymph. Since glucose is converted into glycogen in both fat body and muscle for storage, we measured the level of glycogen in *cg>Dp-RNAi* and in *Mef2>Dp-RNAi*. As shown in *Figure 6—figure supplement 1F–G*, there were no major defects in glycogen storage normalized to protein content in *cg>Dp-RNAi* and *Mef2>Dp-RNAi* compared to control animals fed either a control or a high sugar diet.

As mentioned above, Dp depletion in fat body leads to the appearance of binucleated cells (*Figure 2A–B*; *Guarner et al., 2017*). Therefore, we asked whether this phenotype can be suppressed by food composition. Binucleated cells were readily found in both *cg>Dp-RNAi* and *Mef2>Dp-RNAi* larvae fed a control diet, but the phenotype was not modified by rearing larvae on

high sugar diet (*Figure 6—figure supplement 1H–I*). Since dietary changes suppress the defects and fully rescue animal lethality, we infer that the essential function of Dp in fat body is to maintain homeostasis of the circulating sugar trehalose and to regulate fat storage. The fact that binucleated cells persisted in the fat body of rescued animals suggests that the binucleated phenotype is most likely distinct from the metabolic defects.

## Discussion

The experiments described here were prompted by a paradox: if Dp has tissue-specific functions in both the muscle and fat body that are essential for viability, how is it that restoring expression in just one of these tissues can be sufficient to rescue *Dp*-null mutants to viability? We anticipated that *Dp* loss in one of these tissues might cause changes in the other tissue, and/or that the tissue-specific changes might converge through systemic changes. Our data provide important new insights into the consequences of eliminating E2F/Dp regulation in *Drosophila*. Proteomic and metabolomic profiles reveal extensive changes in *Dp*-depleted tissues and show that the loss of E2F/Dp regulation causes a complex combination of tissue-intrinsic changes and systemic changes. The results described here show that E2F/Dp contributes to animal development and viability by preventing both types of defects.

It is clear from the molecular profiles that *Dp* loss causes extensive metabolic changes in both muscle and fat body. The profiles indicate that E2F/Dp participates, in particular, in the regulation of carbohydrate metabolism in both tissues. Such extensive changes to cellular metabolism likely have biological significance in both contexts, but we know that this is certainly true when *Dp* is depleted in the fat body because the lethality of *cg>Dp-RNAi* could be rescued by increasing dietary sugar. In *Drosophila*, the fat body functions as a key sensor of the nutritional status and it couples systemic growth and metabolism with nutritional availability. The lethality associated with *cg>Dp-RNAi* may, therefore, stem from an imbalance between energy production and systemic growth.

Carbohydrate metabolism defects in *cg>Dp-RNAi* larvae are consistent with studies of mammalian E2F1 in liver and adipose tissue, tissues that metabolize nutrients and stores reserves of lipid and glycogen. E2F1 was shown to be required for the regulation of glycolysis and lipid synthesis in hepatocytes (*Denechaud et al., 2016*). In adipose tissue, E2F1 was implicated in the regulation of PPARγ, the master adipogenic factor, during early stages of adipogenesis (*Fajas et al., 2002*). Moreover, the gene encoding for the pyruvate dehydrogenase kinase 4, a key nutrient sensor and modulator of glucose oxidation, and 6-phosphofructo-2-kinase/fructose-2,6-bisphosphatase, a glycolytic enzyme, are directly regulated by E2F1 (*Fernández de Mattos et al., 2002*; *Hsieh et al., 2008*). Our results highlight the conserved role of E2F/DP in regulating metabolism. A major function for the fat body is to control whole-animal sugar homeostasis. Interestingly, trehalose and glucose metabolism plays a key role in regulating systemic energy homeostasis (*Matsuda et al., 2015*; *Yasugi et al., 2017*). Proper trehalose homeostasis is required for adapting animals to various dietary conditions, as evidenced by *Tps1* mutants, which do not survive in low sugar diet (*Matsuda et al., 2015*). The fact that the lethality associated with *cg>Dp-RNAi* can be suppressed by a sugar supplement that results in a trehalose increase, underscores the importance of E2F function for animal development, which was not previously appreciated.

In contrast with *cg>Dp-RNAi*, the lethality associated with *Mef2>Dp-RNAi* was not rescued by altering the diet. However, the appearance of binucleated cells in the fat bodies of *Mef2>Dp-RNAi* larvae demonstrates that the muscle-specific knockdown of *Dp* does have systemic effects. These binucleated cells resemble defects triggered by the loss of E2F in the fat body and are also seen in *Dp* mutant larvae. We note that the mechanisms leading to the formation of binucleates in *Mef2>Dp-RNAi* animals may differ from *cg>Dp-RNAi* and *Dp-/-* mutants (*Guarner et al., 2017*) since they are not associated with a DNA damage response. Indeed, the binucleated cells in the fat body of *Mef2>Dp-RNAi* animals may be a symptom of stress; evidence for paracrine signals from muscle support this idea (*Demontis et al., 2014*; *Demontis and Perrimon, 2009*; *Zhao and Karpac, 2017*). We infer that the tissue-specific depletion of *Dp* causes systemic effects in both *cg>Dp-RNAi* and *Mef2>Dp-RNAi* larvae, but it is particularly evident in *cg>Dp-RNAi* animals, which have a low level of circulating trehalose in hemolymph.

The level of circulating trehalose is a net result of the amount of trehalose released to hemolymph by fat body and its consumption by other organs, including rapidly growing muscles. The trehalose

requirement of wild-type organs in *cg>Dp-RNAi* animals may provide an additional challenge to the *Dp*-deficient fat body to maintain trehalose homeostasis. Conversely, the reduced demand for trehalose by *Dp*-deficient organs in the *Dp-/-* mutants may shift the balance toward increased levels of circulating trehalose. Thus, one implication of our work is that the severity of the phenotype may differ between the whole-body *Dp-/-* mutant and the tissue-specific *Dp* inactivation.

There is growing evidence of the systemic effects that muscles exert in different settings, including development and aging (*Demontis et al., 2014*; *Demontis et al., 2013*; *Demontis and Perrimon, 2009*; *Zhao and Karpac, 2017*). Myokines are thought to be the main mediators of the inter-tissue communication between muscles and fat body, and other distant tissues. Here, we report that *Dp*-deficient muscle causes the appearance of binucleated cells in the distant fat body of *Mef2>Dp-RNAi* larvae. Our data strongly argue that the fat body phenotype is due to the tissue-specific loss of *Dp* in muscles as the levels of Dp remain unchanged in *Mef2>Dp-RNAi* fat body compared to control animals. This is further validated by lineage tracing, thus confirming the muscle-specific expression of the *Mef2-GAL4* driver. The *Mef2>Dp-RNAi* animals raised on high sugar diet may have a very mild defect, yet not statistically significant, in the formation of lipid droplets in fat body without an effect on global body triglyceride content. Maintaining proper homeostasis of triglyceride metabolism at an organismal level requires the orchestration of elaborated endocrine mechanism for inter-tissue communication (*Heier and Kühnlein, 2018*). Myokines and other cytokines communicate systemic changes in nutrient sensing status and energy substrate storage via cross talk with insulin and Akt signaling pathway (*Demontis et al., 2013*; *Géminard et al., 2009*; *Zhao and Karpac, 2017*).

Since E2F/DP proteins are master regulators of cell proliferation, the finding that *Dp* mutant animals develop to the late pupal stages and largely without defects was surprising. It also gave the impression that E2F/DP could be eliminated without major consequences. The results described here paint a different picture. The molecular profiles show extensive changes in *Dp*-deficient tissues. Indeed, 20% and 51% of the proteins that we quantified in fat bodies and muscles, respectively, were expressed at statistically different levels following tissue-specific *Dp* knockdown. The second conclusion of our work is that Dp inactivation exerts a systemic effect and may impact distant wild-type tissues, which was completely missed in previous analysis of whole-body *Dp* mutants and became only evident when *Dp* was inactivated in a tissue-specific manner. We report that *Dp*-deficient muscles induce mild defects in distant fat body, while downregulation of Dp in fat body results in low level of circulating trehalose, which is likely the main cause of lethality of *cg>Dp-RNAi* animals. Thus, the *Dp* mutant phenotype is a complex combination of both tissue-intrinsic and systemic effects.

# Materials and methods

**Key resources table**

| Reagent type (species) or resource | Designation | Source or reference | Identifiers | Additional information |
|---|---|---|---|---|
| Gene (*Drosophila melanogaster*) | DP transcription factor; Dp | NA | Flybase:FBgn0011763 | |
| Genetic reagent (*Drosophila melanogaster*) | *UAS-Dp-RNAi* | Vienna *Drosophila* Resource Center | Flybase:FBst0450633 | |
| Genetic reagent (*Drosophila melanogaster*) | *Dp[a3]* | PMID:15798191, 9271122 | | |
| Genetic reagent (*Drosophila melanogaster*) | *Dp[a4]* | PMID:15798191, 9271122 | | |
| Genetic reagent (*Drosophila melanogaster*) | *Df(2R)Exel7124* | PMID:15798191, 9271122 | Flybase:FBab0038034; RRID:BDSC_7872 | |
| Genetic reagent (*Drosophila melanogaster*) | *UAS-Dp* | PMID:8670872; 9657151; 26823289 | | |
| Genetic reagent (*Drosophila melanogaster*) | *Dp[GFP]; P{PTT-GA}DpCA06954* | PMID:17194782 | | |

*Continued on next page*

*Continued*

| Reagent type (species) or resource | Designation | Source or reference | Identifiers | Additional information |
|---|---|---|---|---|
| Genetic reagent (*Drosophila melanogaster*) | P{Cg-GAL4.A}2 | Bloomington *Drosophila* Stock Center | Flybase:FBti0027802; RRID:BDSC_7011 | |
| Genetic reagent (*Drosophila melanogaster*) | P{GAL4-Mef2.R}3 | Bloomington *Drosophila* Stock Center | Flybase:FBst0027390; RRID:BDSC_27390 | |
| Genetic reagent (*Drosophila melanogaster*) | P{VALIUM20-mCherry}attP2 | Bloomington *Drosophila* Stock Center | Flybase:FBti0143385; RRID:BDSC_35785 | |
| Genetic reagent (*Drosophila melanogaster*) | UAS-Luc-RNAi ; P{y[+t7.7] v[+t1.8]=TRiP.JF01355}attP2 | Bloomington *Drosophila* Stock Center | Flybase:FBti0130444; RRID:BDSC_31603 | |
| Genetic reagent (*Drosophila melanogaster*) | r4-GAL4 | Bloomington *Drosophila* Stock Center | Flybase:FBti0132496; RRID:BDSC_33832 | |
| Genetic reagent (*Drosophila melanogaster*) | P{Lpp-GAL4.B}3 | Bloomington *Drosophila* Stock Center | Flybase:FBti0185670; RRID:BDSC_84317 | |
| Genetic reagent (*Drosophila melanogaster*) | G-TRACE | *Evans et al., 2009* | | |
| Genetic reagent (*Drosophila melanogaster*) | Canton-S | Bloomington *Drosophila* Stock Center | Flybase:FBst0064349 | |
| Genetic reagent (*Drosophila melanogaster*) | w$^{1118}$ | Bloomington *Drosophila* Stock Center | FLYB:FBst0003605; RRID:BDSC_3605 | |
| Antibody | Mouse monoclonal anti-dDP (Yun6) | PMID:8670872 | RRID:AB_2889822 | (1:10) |
| Antibody | Rabbit polyclonal anti-Dp antibodies (#212) | PMID:12975318 | NA | (1:300) |
| Antibody | FITC Goat polyclonal anti-GFP (ab6662) | Abcam | RRID:AB_305635 | (1:500) |
| Antibody | Guinea pig anti-Rad50 | PMID:19520832 | NA | (1:100) |
| Antibody | Guinea pig anti-Mre11 | PMID:19520832 | NA | (1:100) |
| Peptide, recombinant protein | Porcine trehalase | Sigma | T8778-1UN | |
| Peptide, recombinant protein | Amyloglucosidase | Sigma | A1602 | |
| Peptide, recombinant protein | Triglyceride reagent | Sigma | T2449 | |
| Commercial assay or kit | Glucose (HK) assay reagent | Sigma | G3293 | |
| Chemical compound, drug | Triglyceride reagent | Sigma | T2449 | |
| Chemical compound, drug | Trehalose | Sigma | 90208 | |

*Continued on next page*

*Continued*

| Reagent type (species) or resource | Designation | Source or reference | Identifiers | Additional information |
|---|---|---|---|---|
| Chemical compound, drug | Glucose standard solutions | Sigma | G3285 | |
| Chemical compound, drug | Free glycerol reagent | Sigma | F6428 | |
| Chemical compound, drug | BODIPY 493/503 | Invitrogen | D3922 | 0.5 µg/ml |
| Chemical compound, drug | Bradford standard assay | Bio-Rad | 500–0006 | |
| Software, algorithm | ImageJ 1.52k5 | National Institutes of Health, Bethesda, MD https://imagej.nih.gov/ij/ | RRID:SCR_003070 | |
| Software, algorithm | Photoshop CC 2019 | Adobe Systems | RRID:SCR_014199 | |
| Software, algorithm | Functional Annotation Clustering – DAVID platform | https://david.ncifcrf.gov/summary.jsp | RRID:SCR_001881 | |
| Software, algorithm | Metaboanalyst | http://www.metaboanalyst.ca | RRID:SCR_015539 | |
| Software, algorithm | GraphPad Prism version 9.0.1 | GraphPad Software | RRID:SCR_002798 | |
| Software, algorithm | R | R Project for Statistical Computing | RRID:SCR_001905 | |

## Fly stocks

Flies were raised in vials containing standard cornmeal agar medium at 25° C. The $w^{1118}$ flies were used as wild-type control flies. Either the $Dp^{a3}$ and $Dp^{a4}$ alleles or $Dp^{a3}$ and deficiency $Df(2R)$ *Exel7124*, which deletes the entire Dp gene, were used in this work to obtain the trans-heterozygous Dp mutant larvae for proteome and metabolome experiments, respectively (*Frolov et al., 2005*; *Royzman et al., 1997*). The trans-heterozygous $E2f2^{76Q1/c03344}$; $E2f1^{91/m729}$ mutant animals were used as double *E2f1* and *E2f2* mutants. The GAL4 drivers, P{Cg-GAL4.A}2, P{w[+mC]=Lpp-GAL4.B}3 and P{GAL4-Mef2.R}3, and the following control UAS-RNAi lines from the TRIP collection: *Luciferase* (P{y[+t7.7] v[+t1.8]=TRiP.JF01355}attP2) and *mCherry* (P{VALIUM20-mCherry}attP2) obtained from Bloomington *Drosophila* Stock Center (Bloomington, IN). The line *UAS-Dp-RNAi* was obtained from the library RNAi-GD (ID 12722) at the Vienna *Drosophila* Resource Center (Vienna, Austria). The stock *UAS-G-TRACE* (*Evans et al., 2009*) was used to trace the expression of the drivers. The P{PTT-GA}Dp$^{CA06954}$ line from the Carnegie collection (*Buszczak et al., 2007*), here annotated as DpGFP, contains a GFP-expressing protein trap insertion (*Zappia and Frolov, 2016*). The P{UAS-Dp.D} was used to overexpress Dp in the Dp mutant background (*Du et al., 1996*; *Neufeld et al., 1998*; *Zappia and Frolov, 2016*).

## Fly viability assay

The total number of pupae, pharate pupae, and adult flies able to eclose out of the pupal case were scored. The pupal developmental stages were assessed by following markers of metamorphosis (*Ashburner et al., 2005*). At least 58 flies per group were scored in a minimum of three independent experiments.

## Fly food recipes

All flies were raised on Bloomington standard cornmeal food. After eclosion, adults were transferred to different fly food composition, which was made based on the semi-defined control diet (*Lee and Micchelli, 2013*) with adjustments. Control diet was made of 1% agar (Lab Scientific, Fly 8020), 4.35% brewers yeast (MP Biomedicals, 2903312), 0.04% lecithin (soybean, MP Biomedicals, 102147), propionic acid (0.5% v/v), and 7.9% sucrose (FCC Food grade, MP Biomedicals, 904713). The adjustments for each food type are detailed in *Supplementary file 8*.

## Hemolymph extraction

Hemolymph was collected from ~10 or 15 third instar larvae per sample. Protocol was adapted from *Tennessen et al., 2014*. Each animal was rinsed with ddH$_2$O, wipped to remove excess of water, carefully punctured in the mouth hook using a tungsten needle and placed in a 0.5 ml tube with a hole at the bottom of the tube. This tube was then placed in a 1.5 ml tube and centrifuged to maximum speed for 10 s. Approximately 1 µl of hemolymph was collected for each sample. Hemolymph was diluted 1:50 in trehalase buffer (TB) (5 mM Tris pH 7.6, 137 mM NaCl, 2.7 mM KCl). Samples were heat-treated for 5 min at 70℃ and centrifuge for 3 min at maximum speed at 4℃. Supernatant was quickly snap-frozen and stored at −80℃ until all samples were harvested.

## Trehalose measurement

Hemolymph samples were further diluted to final dilution 1:150 with buffer TB. Circulating trehalose was measured in hemolymph as previously described (*Tennessen et al., 2014*). Briefly, an aliquot of each sample was treated with porcine trehalase (Sigma, T8778-1UN) overnight at 37℃ in a G1000 Thermal cycler (Bio-Rad) to digest trehalose and produce free glucose. In parallel, another aliquot was incubated with buffer TB to determine the levels of glucose. The total amount of glucose was determined using the glucose (HK) assay reagent (Sigma, G3293) following a 15 min incubation at room temperature. Trehalose (Sigma, 90208) and glucose standard solutions (Sigma, G3285) were used as standards. Plate reader BioTek Epoch was used to read absorbance at 340 mm. The trehalose concentrations for each sample were determined by subtracting the values of free glucose in the untreated samples. Each sample was measured twice, a total of six independent biological samples were collected by group, and three independent experiments were done.

## Quantification of glycogen and protein content

Third instar larvae were harvested to measure glycogen as previously described (*Tennessen et al., 2014*). Briefly, seven animals were collected per sample, rinsed with ddH$_2$O, and homogenized in 100 µl PBS 1×. Samples were heat-treated at 70℃ for 10 min and centrifuged at maximum speed for 3 min at 4℃. Supernatant was stored at −80℃ until all samples were collected. Samples were diluted 1:6 in PBS 1× for the assay and transferred to two wells. One well was treated with amyloglucosidase (Sigma A1602) and the second well with PBS 1×. The plate was incubated at 37℃ for 1 hr. Then, the total amount of glucose was determined using 100 µl of glucose (HK) assay reagent (Sigma, G3293) following a 15 min incubation at room temperature. Glycogen and glucose standard solutions were used as standards. Plate reader BioTek Epoch was used to read absorbance at 340 mm. The glycogen concentrations for each sample were determined by subtracting the values of free glucose in the untreated samples. Each sample was measured twice, a total of six independent biological samples were collected by group, and three independent experiments were done.

Total glycogen was normalized to soluble protein amount. Aliquots of larval homogenate were removed prior heat treatment to measure soluble protein using a Bradford assay (Bio-Rad 500–0006) with BSA standard curves.

## Triglycerides quantification

A coupled colorimetric assay was used to quantify triglycerides by measuring free glycerol as previously described (*Tennessen et al., 2014*). Briefly, seven animals were collected per sample, rinsed with ddH$_2$O and homogenized in cold 100 µl PBS-T (PBS 0.05% Tween-20). Samples were heat-treated at 70℃ for 10 min and stored at −80℃ until all samples were collected. Samples were diluted 1:6 in PBS-T for the assay and transferred to two wells. One well was treated with triglyceride reagent (Sigma, T2449) and the second well with PBS-T. The plate was incubated at 37℃ for 30 min

in a G1000 Thermal cycler (Bio-Rad). Then, the total amount of free glycerol was determined using 100 µl free glycerol reagent (Sigma, F6428) following a 5 min incubation at 37°C. Glycerol (triolein, Sigma, G7793) standard solution was used as standard. Plate reader BioTek Epoch was used to read absorbance at 540 mm. The triglyceride concentrations for each sample were determined by subtracting the values of free glycerol in the untreated samples. Each sample was measured twice, a total of six independent biological samples were collected by group, and three independent experiments were done. Total triglyceride was normalized to soluble protein amount as described above.

## Immunofluorescence

Tissues were dissected and fixed in 4% formaldehyde in PBS for 30 min. Then, tissues were permeabilized during 10 or 15 min in 0.1% Triton X-100 in PBS or in 0.3% Triton X-100 in PBS for muscles tissues. Tissues were washed and blocked in 1% or 2% BSA PBS. Primary antibodies were incubated overnight at 4°C in 2% BSA and 0.1% Triton X-100 in PBS. After washing three or four times for 10 min each in 0.1% Triton X-100 (in PBS), secondary antibodies (Alexa Fluor, Cy3- or Cy5-conjugated anti-mouse and anti-rabbit secondary antibodies, Life Technologies and Jackson Immunoresearch Laboratories) were incubated for 60 or 90 min in 10% normal goat serum 0.1% Triton X-100 in PBS. After washing three times with 0.1% Triton X-100 (in PBS), tissues were mounted on glass slides in glycerol with antifade or in Vectashield with DAPI (Vector Laboratories). All steps were performed at room temperature, unless otherwise stated.

In the case of fat bodies, the fixation was done for 60 min and PBS 1× was used throughout the protocol instead of 0.1% Triton X-100 in PBS 1× (*Guarner et al., 2017*). For larval body wall musculature staining, larva was dorsally opened, pinned in a Sylgard dish, and fixed for 20 min. A minimum of five to eight animals per genotype was dissected per experiment, and the staining was carried out two or three times.

The primary antibodies were mouse monoclonal anti-Dp antibody (Yun6, dil 1:10, *Du et al., 1996*) used in fat bodies and rabbit polyclonal anti-Dp antibodies (#212, dil 1:300 *Dimova et al., 2003*) used in muscles, anti-GFP (FITC, 1:500, Abcam ab6662), Guinea pig anti-Rad50 (1:100, *Gao et al., 2009*), Guinea pig anti-Mre11 (1:100, *Gao et al., 2009*). Rhodamine–phalloidin or fluorescein isothiocyanate–phalloidin were used to counterstain, and DAPI for nucleus staining.

## Lipid droplet detection

To visualize lipid droplets, dissected third instar larval fat bodies were fixed in 4% formaldehyde in PBS for 1 hr at room temperature and washed three times in PBS 1×. Fat bodies were incubated in solution containing both 0.5 µg/ml BODIPY 493/503 (Invitrogen, D3922) and DAPI diluted in PBS 1×, for 10 min at room temperature, then washed three times in PBS 1×.

## Confocal microscopy/image acquisition

Fluorescent images were acquired with the laser scanning confocal microscope (Zeiss LSM 700) using × 20/0.8, and × 40/1.2 objectives at University of Illinois at Chicago and 710 Zeiss Confocal microscope at MGH Cancer Center. Images were processed using ImageJ (1.52k5, National Institutes of Health, Bethesda, MD) and Photoshop CC 2019 (Adobe Systems). All images are confocal single plane images, except otherwise stated. Only representative images are shown.

## Quantitative proteomics

### Sample preparation

Fat bodies and thoracic muscles were dissected in cold PBS 1×. To pellet the dissected tissues, vials were centrifuged at 4°C at maximum speed, and PBS was removed prior to snap-freezing. Collected tissues were thaw and resuspended in modified protein lysis buffer (50 mM HEPES pH 8, 100 mM KCl, 2 mM EDTA, 10 mM NaF, 10% glycerol, 0.1% NP-40, 1 mM dithiothreitol, 1 mM PMSF, and Roche protease inhibitors) and homogenized on ice. The amount of total protein was measured with Lowry colorimetric assay (DC, Bio-Rad) for fat bodies and Bradford standard assay (Bio-Rad 500–0006) for muscles. Western blotting was carried out using standard procedures. The mouse anti-DP Yun (#6, 1:5 *Du et al., 1996*) and the mouse beta-actin (1:1000, Abcam, Cat# ab8224) antibody were used as loading control in western blot assays.

## Multiplexed quantitative mass spectrometry-based proteome

The TMT-10 plex reagents and the simultaneous precursor selection (SPS)-MS3 method on an Orbitrap Fusion mass spectrometer (Thermo Scientific) (*Edwards and Haas, 2016*; *Guarner et al., 2017*; *Lapek et al., 2017*; *McAlister et al., 2014*) were used to profile $w^{1118}$ and Dp-/- whole larval lysates, and *Mef2>mCherry-RNAi* and *Mef2>Dp-RNAi* thoracic muscle lysates in triplicate and quadruplet, respectively. Disulfide bonds were reduced, free thiols were alkylated with iodoacetamide, proteins were purified by MeOH/CHCl3 precipitation and digested with Lys-C and trypsin, and peptides were labeled with TMT-10plex reagents (Thermo Scientific) (*Edwards and Haas, 2016*; *McAlister et al., 2014*). Labeled peptide mixtures were pooled and fractionated by basic reversed-phase HPLC. Four fractions were analyzed by multiplexed quantitative proteomics performed on an Orbitrap Fusion mass spectrometer (Thermo Scientifc) using an SPS-based MS3 method (*McAlister et al., 2014*). MS2 spectra were assigned using a SEQUEST-based proteomics analysis platform (*Huttlin et al., 2010*). The protein sequence database for matching the MS2 spectra was based on v5.57 of the *Drosophila* melanogaster proteome retrieved from Flybase (*Attrill et al., 2016*). Peptide and protein assignments were filtered to an FDR of <1% employing the target-decoy database search strategy (*Elias and Gygi, 2007*) and using linear discriminant analysis and posterior error histogram sorting (*Huttlin et al., 2010*). Peptides with sequences contained in more than one protein sequence from the UniProt database were assigned to the protein with most matching peptides (*Huttlin et al., 2010*). We extracted TMT reporter ion intensities as those of the most intense ions within a 0.03 Th window around the predicted reporter ion intensities in the collected MS3 spectra. Only MS3 with an average signal-to-noise value of larger than 20 per reporter ion as well as with an isolation specificity (*Ting et al., 2011*) of larger than 0.75 were considered for quantification. A two-step normalization of the protein TMT intensities was performed by first normalizing the protein intensities over all acquired TMT channels for each protein based on the median average protein intensity calculated for all proteins. To correct for slight mixing errors of the peptide mixture from each sample, a median of the normalized intensities was calculated from all protein intensities in each TMT channel and the protein intensities were normalized to the median value of these median intensities.

## Proteomics analysis

A total of 6578 and 5730 proteins in fat body and muscles, respectively, were quantified across all experimental conditions. Only shared proteins between fat bodies and muscles were selected for an integrative analysis, all downstream analysis were done on combined 5024 proteins. Differential protein expressions between $w^{1118}$ and Dp-/- fat bodies and between *Mef2>mCherry-RNAi* and *Mef2>Dp-RNAi* proteomes were calculated using a moderated *t*-test. The Benjamini-Hochberg multiple hypothesis correction was applied to calculate corrected p-values (FDR). Differential expression of proteins was considered significant with an FDR < 5% and an absolute fold change greater than 1.5.

## Functional enrichment analysis

Functional annotation clustering of the differentially expressed proteins were analyzed using DAVID platform (https://david.ncifcrf.gov/summary.jsp *Huang et al., 2009a*; *Huang et al., 2009b*). Functional terms related to biological process, KEGG, and UP_Keyword were identified using FDR < 0.05 in the selected top cluster. Furthermore, gene ontology enrichment was analyzed by selecting KEGG pathways and using FDR < 5%.

## **Metabolomics profiles**

### Pre-extraction of metabolites from whole larvae and $^{13}C$ labeling of whole larvae

Six biological samples per genotype were processed exactly as previously published (*Nicolay et al., 2013*). From each of the vials that contained ~30 larvae, animals were isolated and washed twice in ddH2O to remove any excess foodstuff, outside unlabeled metabolites, or excess $^{13}C$ labeled glutamine. Then animals were collected in 1.5 ml tubes and the total weight of each collection of starting material was ~10 mg to achieve detection of unstable metabolites. Approximately 3 mg of starting material was sufficient for most metabolites. For each condition tested, metabolites were extracted

from six to eight biological replicates of pooled animals from each genotype. Samples were then snap-frozen in liquid nitrogen and either stored at −80℃ for further processing or processed immediately.

Snap-frozen samples were kept on dry ice during extraction; 500 µl of −80℃ MeOH:$H_2O$ (80:20) was added to each pellet. Pellets were homogenized by hand with a pestle using three to five strokes. Samples were vortexed at 4℃ for 1 min and left at −80℃ for 4 hr. After 4 hr, samples were vortexed at 4℃ for 30 s. Samples were clarified at 20 K × $g$, for 0.25 hr at 4℃. Clarified supernatant was transferred to a new 1.5 ml tube and stored at −80℃. Each pellet was re-extracted with −80℃ MeOH:$H_2O$ (80:20), vortexed for 30 s at 4℃, and stored at −80℃ for 0.5 hr. Re-extracted material was vortexed for 30 s at 4℃ and then clarified at 20 K × $g$, for 0.25 hr at 4℃. Clarified supernatants were combined and clarified one more time. Combined supernatants were then evaporated by SpeedVac, snap-frozen in liquid nitrogen, and stored at −80℃. Prior to mass spectrometry analysis, samples were resuspended using 20 µl HPLC grade water.

### LC-MS/MS

LC-MS methodology was performed as described in *Nicolay et al., 2015*. In brief, nanospray HPLC-MS was carried out with an Agilent 1260 Infinity pump coupled to a FAMOS+ autosampler and an Exactive Orbitrap mass spectrometer. The mass spectrometer was equipped with an electrospray ionization source operated in negative mode. The mass spectrometer was calibrated using a negative ion calibration solution (Pierce 88324) and the optimized conditions were spray voltage 1.8 kV, spray current 2.1 µA, capillary temperature 301℃, capillary voltage −52.5, tube lens voltage −150, skimmer voltage −42. The mass spectrometer was run in full scan mode (80–1000 m/z range) with an R = 100,000 at 1 Hz (1 scan/s) with the use of the ion pairing reagent, Tributylamine (Sigma 471313). The stationary phase was a C18 medium (3 µm, 200 A) from Maccel. The LC method used was as follows: 0 min, 0% B; 11 min, 5% B; 24 min, 100% B; 30 min, 100% B; 31 min, 0% B, 40 min, 0% B. Injection volume was 1 µl. Flow rate at column bed was 400 nl/min. Buffer A: 5% methanol, 10 mM TBA, 10 mM acetic acid. Buffer B: Methanol. Raw data files were transformed and analyzed in MAVEN (*Clasquin et al., 2012*; *Melamud et al., 2010*).

### Metabolomics analysis

For analyses of metabolite pools, the free portal Metaboanalyst (http://www.metaboanalyst.ca) was used. Raw metabolite measurement data were converted to achieve a normal distribution of the data. For each metabolite, data were median-centered, then $log_2$-transformed across the genotypes followed by autoscaling. Among the 258 metabolites, only 210 had at least three strong peaks (out of the 6–8 per group) in each of the genotypes. Analysis was done using data from the 210 metabolites that gave reproducible peaks above noise. Standard compound names were used and the compound pathway library was *Drosophila melanogaster*. Following normalization, altered metabolite levels were functionally compared across all three genotypes. Only metabolites that were significantly altered in the *Dp-/-* mutant when compared to both the Canton S and $w^{1118}$ control genotypes were considered in the analysis to account for the variation on genetic background. KEGG enrichment analysis was done using metabolites changed up and down in *Dp-/-* mutant to determine what metabolic pathways were altered. Direct comparisons between normalized values of specific metabolites were done using Excel and significance was tested using one-way ANOVA followed by Tukey's post hoc analysis.

## Quantification and statistical analysis

### Image analysis

Muscle area in body wall muscles VL3 was measured for each animal using ImageJ. Raw data values, ANOVA results, multiple comparisons tests, and summary statistics are included in the associated Source File files.

The size of lipid droplets was quantified using the 'analyze particles' function of ImageJ.

The number of binucleated cells in fat bodies was manually scored on the microscope and normalized to the total number of nuclei counted per field. Raw data values, ANOVA results, multiple comparisons tests, and summary statistics are included in the associated Source File files.

The ratio of Dp (and Dp$^{GFP}$) signal relative to nuclear area was calculated using Fiji (https://fiji.sc/) in fat bodies and ImageJ (https://imagej.nih.gov/ij/) in muscles. Measurements for both parameters (raw intensity and nuclear area) were collected simultaneously in individual images.

Scripts used in ImageJ for automatic and unbiased quantification of lipid droplets are included in Source Data. All statistics and graphs were generated with the GraphPad Prism version 9.0.1 (GraphPad Software). The group means were analyzed for overall statistical significance using non-parametric test, including Kruskal-Wallis test followed by Dunn's multiple comparisons test and Mann-Whitney test, and two-way ANOVA followed by Tukey's multiple comparisons test in control and high sugar diet experiments. Both a Spearman's test for heteroscedasticity and a Kolmogorov-Smirnov and a Shapiro-Wilk (W) test for normality were assessed before choosing two-way ANOVA statistical analysis. Details on the sample size, number of independent experiment, and statistical analysis are listed in figure legends.

Raw data values, ANOVA results, multiple comparisons tests, and summary statistics are included in the associated Source File files for *Figures 1*, *4,* and *6*.

## Proteomics and metabolomics analysis
All plots for proteomic and metabolomic were generated using R. Further details on the analysis can be found in the proteome and metabolome sections in Materials and methods.

## Acknowledgements

We thank Kristin White, Ron Dubreuil, and Jason Tennessen for helpful discussions, Isabel Liseth for technical help. We are grateful to the Bloomington *Drosophila* Stock Center (supported by NIH grant P40OD018537), the Vienna *Drosophila* Resource Center, the TRiP at Harvard Medical School for fly stocks, the Developmental Studies Hybridoma Bank (DSHB) for antibodies, and to Flybase for online resources on the Database of *Drosophila* Genes and Genomes. This work was supported by NIH grant R35GM131707 (to MVF) and R01GM117413 (to NJD).

## Additional information

### Funding

| Funder | Grant reference number | Author |
| --- | --- | --- |
| National Institute of General Medical Sciences | R35GM131707 | Maxim V Frolov |
| National Institute of General Medical Sciences | R01GM117413 | Nicholas J Dyson |

The funders had no role in study design, data collection and interpretation, or the decision to submit the work for publication.

### Author contributions
Maria Paula Zappia, Conceptualization, Formal analysis, Investigation, Methodology, Writing - original draft, Writing - review and editing; Ana Guarner, Conceptualization, Formal analysis, Investigation, Methodology, Writing - original draft; Nadia Kellie-Smith, Alice Rogers, Brandon Nicolay, Investigation; Robert Morris, Brandon Nicolay, Myriam Boukhali, Formal analysis, Methodology; Wilhelm Haas, Formal analysis, Supervision, Methodology; Nicholas J Dyson, Conceptualization, Supervision, Funding acquisition, Writing - review and editing; Maxim V Frolov, Conceptualization, Supervision, Funding acquisition, Project administration, Writing - review and editing

### Author ORCIDs
Maria Paula Zappia 🆔 https://orcid.org/0000-0003-0449-5445
Maxim V Frolov 🆔 https://orcid.org/0000-0003-3953-3739

### Decision letter and Author response
Decision letter https://doi.org/10.7554/eLife.67753.sa1

Author response https://doi.org/10.7554/eLife.67753.sa2

# Additional files

## Supplementary files

• Source data 1. MacroScript used to analyze particle size or LipidDroplets in ImageJ.

• Supplementary file 1. Proteins identified and quantified by tandem-mass tag-mass spectrometry (TMT-MS) normalized intensities for 6580 proteins in fat body and 5730 in muscles. Third instar larval $w^{1118}$ and *Dp-/-* were collected. Fat body was dissected in three replicates for each genotype. *Mef2>mCherry-RNAi* and *Mef2>Dp-RNAi* were collected at pharate stage. Thoracic muscles were dissected in four replicates for each genotype.

• Supplementary file 2. Proteomic effect of E2F/Dp loss in larval fat body and thoracic muscles. Tandem-mass tag-mass spectrometry (TMT-MS) data were log transformed and fold change was calculated between E2f/Dp-depleted tissue and control. Only 5024 proteins detected in both tissue are displayed here. Third instar larval fat body from $w^{1118}$ and *Dp-/-* was collected. *Mef2>mCherry-RNAi* and *Mef2>Dp-RNAi* were collected at pharate stage and thoracic muscles were dissected

• Supplementary file 3. Enrichment analysis for KEEG pathways using the functional annotation analysis in DAVID. List of upregulated and downregulated proteins in fat body and muscles were selected for enrichment analysis using cutoff FDR<5%, and FC>1.5 and FC<1.5, respectively.

• Supplementary file 4. Functional annotation clustering for common changes between fat body and muscle. Upregulated and downregulated proteins that are in common between fat body and muscle were selected for DAVID functional annotation tool. Annotation terms were clustered based on similarity.

• Supplementary file 5. Raw values for metabolomics dataset. Third instar larval *Dp-/-* mutant and controls, Canton-S and $w^{1118}$, were profiled by liquid chromatography tandem mass spectrometry (LC-MS/MS). Original and preprocessing data. N/A means no detectable peak was found for that metabolite in that sample.

• Supplementary file 6. Normalized values for only significantly altered metabolites between *Dp-/-* mutant and two controls, Canton-S and $w^{1118}$. Third instar larval *Dp-/-* mutant and controls, Canton-S and $w^{1118}$, were profiled by liquid chromatography tandem mass spectrometry (LC-MS/MS). Data were normalized, median centered by metabolite, and scaled. Only metabolites that were significantly altered in the Dp-/- mutant when compared to both the Canton S and $w^{1118}$ genotypes are displayed.

• Supplementary file 7. Overrepresented KEGG pathways from altered metabolites in Dp mutant. Enrichment analysis of the KEGG metabolic pathways.

• Supplementary file 8. Fly food medium composition semi-defined fly food was adapted from *Lee and Micchelli, 2013*.

• Transparent reporting form

## Data availability

All mass spectrometer RAW files for quantitative proteomics analysis can be accessed through the MassIVE data repository (https://massive.ucsd.edu/ProteoSAFe/static/massive.jsp) under the accession number MSV000086854.

The following dataset was generated:

| Author(s) | Year | Dataset title | Dataset URL | Database and Identifier |
|---|---|---|---|---|
| Zappia MP, Guarner A, Morris R, Boukhali M, Haas W, Dyson NJ, Frolov MV | 2021 | The impact of E2F/Dp inactivation on metabolism differs between muscles and fat body cells | ftp://massive.ucsd.edu/MSV000086854/ | MassIVE data repository (massive.ucsd.edu), MSV000086854 |

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
