## [Decision Letter]

**Acceptance summary:**

The paper's principal conclusion is that, apart from its well-known functions in controlling cell cycle progression, E2F/Dp activity regulates carbohydrate metabolism, and this function is essential for normal development. The findings are significant in that they broaden our understanding of this important growth-regulatory transcription factor.

**Decision letter after peer review:**

Thank you for submitting your article "E2F/Dp inactivation in fat body cells triggers systemic metabolic changes" for consideration by *eLife*. Your article has been reviewed by 2 peer reviewers, and the evaluation has been overseen by a Reviewing Editor and K VijayRaghavan as the Senior Editor. The following individual involved in review of your submission has agreed to reveal their identity: Bruce A Edgar (Reviewer #1).

Essential revisions:

1) The 4 points raised by reviewer 2 in their public review should be addressed.

*Reviewer #1 (Recommendations for the authors):*

Although I found this paper ingenious and rather interesting, the mechanistic basis of the carbohydrate defects presented is not really addressed directly, and so the defects remain a phenotype of undefined origin. We can only assume that, somehow, E2F/Dp complexes are re-targeted away from cell cycle genes and towards carbohydrate metabolism genes in late-stage (pupal) fat body. Knowing whether this was the case, and what the mechanism of retargeting is, could be quite interesting. RNAseq experiments, combined with ATACseq, ChIP, or DamID experiments to map Dp binding sites on chromatin, could address this. Given that E2F/Dp complexes are transcription factors, it is surprising that such experiments are not presented here. It is also surprising that the probably earlier role of E2F in fat body and muscle cell division and DNA endoreplication is not discussed. Clearly, the decision to use proteomics was effective for understanding the Dp mutant phenotype and identifying the root cause of pupal lethality. Nevertheless, I'd like to encourage the authors to provide some data that would illuminate how and why Dp controls this carbohydrate metabolism program in the fat body, at the level of gene transcription. Is there a temporal switch that alters E2F/Dp target genes (from cell cycle control to metabolic control) at some time in development? What gives Dp this unusual spectrum of gene targets in the fat body?

A second question I had is whether Dp loss, in either fat body or muscle, impairs DNA endoreplication, and whether this might be the cause of the poor development and reduced size of these tissues (reduced muscle fiber size is shown). Perhaps these effects are detailed in earlier papers, but if they are, the data are not cited here. If data on DNA ploidy and cell size have not been presented before, they probably should be presented here.

*Reviewer #2 (Recommendations for the authors):*

The study might benefit from authors reconsidering and rephrasing the interpretation or presenting evidence to deepen the current interpretation.

1. Figure 2A, I am unable to discern binucleated nuclei clearly and the authors should show better and higher mag representations? The representations in Sup 2 A seem to be more convincing and of higher clarity. The authors might want to consider doing a similar representation.

2. Rad50 panel in Figure 2C under *Mef2*>Dp-RNAi column is completely black, whereas there is back ground level staining observed in controls. Is this a representative image? Is this because of the plane selected? It would be better to show projection of a few stacks.

3. Figure 3, where they are using GRACE to confirm that developmentally the *Mef2* and cg drivers don't show over lapping expression is an important point. However it adds no new value to the storyline and is largely a well-executed control experiment. I feel it belongs in Supplemental figures. In Figure 3D-H, the color coding is difficult to keep track of amongst the figures and would recommend labeling the x-axis of these figures. This is also an issue in Figure S3B/E/G, where adding the genotype label instead of a color code would make the figures easier to follow.

4. Figure 4, and Figure 2 convey the same message, Dp function in muscles has a systemic effect on Fat body. Hence it seems they belong in the same figure.

5. Scale bars in figures are required.

Text suggestions:

– Lines 102-107 is a run-on sentence that should be broken down.

– Line 174 should reference the figure which is being described.

– Paper is very jargon-heavy and requires a definition for a broader audience. One abbreviation that isn't defined in the paper is APF.

---

## [Author Response]

Reviewer #1 (Recommendations for the authors):Although I found this paper ingenious and rather interesting, the mechanistic basis of the carbohydrate defects presented is not really addressed directly, and so the defects remain a phenotype of undefined origin. We can only assume that, somehow, E2F/Dp complexes are re-targeted away from cell cycle genes and towards carbohydrate metabolism genes in late-stage (pupal) fat body. Knowing whether this was the case, and what the mechanism of retargeting is, could be quite interesting. RNAseq experiments, combined with ATACseq, ChIP, or DamID experiments to map Dp binding sites on chromatin, could address this. Given that E2F/Dp complexes are transcription factors, it is surprising that such experiments are not presented here. It is also surprising that the probably earlier role of E2F in fat body and muscle cell division and DNA endoreplication is not discussed. Clearly, the decision to use proteomics was effective for understanding the Dp mutant phenotype and identifying the root cause of pupal lethality. Nevertheless, I'd like to encourage the authors to provide some data that would illuminate how and why Dp controls this carbohydrate metabolism program in the fat body, at the level of gene transcription. Is there a temporal switch that alters E2F/Dp target genes (from cell cycle control to metabolic control) at some time in development? What gives Dp this unusual spectrum of gene targets in the fat body?

We agree with the reviewer it is indeed a very relevant question. Please see reply to comment in Public Review. We don’t have any data on fat body tissue-specific data for E2F/Dp target genes, but muscle-specific data for both E2F/Dp and Rbf chromatin occupancy were published in Zappia et al. 2019, Cell Reports 26, 702–719. Our hypothesis is that there is no temporal switch since the complex can be found on both group of genes, cell cycle and metabolic genes. We are currently thoroughly investigating how E2F/Dp regulates the expression of some carbohydrate target genes. This is still work in progress.

A second question I had is whether Dp loss, in either fat body or muscle, impairs DNA endoreplication, and whether this might be the cause of the poor development and reduced size of these tissues (reduced muscle fiber size is shown). Perhaps these effects are detailed in earlier papers, but if they are, the data are not cited here. If data on DNA ploidy and cell size have not been presented before, they probably should be presented here.

Endoreplication has not been analyzed in Dp-depleted muscles yet. There have been several attempts to quantify DNA ploidy in Dp-depleted fat body, but the results were inconclusive. However, in previous work it was shown that the DNA copy number of the under-replicated (UR) regions in the WT fat body were increased in the *Dp-/-* mutant fat body by analyzing the copy number variation using next generation DNA sequencing. Thus, loss of Dp in fat body increases replication within regions of the genome that are normally under-replicated (Figure 5F, Guarner et al. Dev Cell 2017).

Reference was added to manuscript on page 6.

Reviewer #2 (Recommendations for the authors):The study might benefit from authors reconsidering and rephrasing the interpretation or presenting evidence to deepen the current interpretation.

We appreciate the comments of the reviewer. Please see points addressed below.

1. Figure 2A, I am unable to discern binucleated nuclei clearly and the authors should show better and higher mag representations? The representations in Sup 2 A seem to be more convincing and of higher clarity. The authors might want to consider doing a similar representation.

New images were taken and left panel was replaced in Figure 2A.

2. Rad50 panel in Figure 2C under Mef2>Dp-RNAi column is completely black, whereas there is back ground level staining observed in controls. Is this a representative image? Is this because of the plane selected? It would be better to show projection of a few stacks.

Image for *Mef2>Dp-RNAi* was replaced in Figure 2C. Background for Rad50 was adjusted accordingly.

3. Figure 3, where they are using GRACE to confirm that developmentally the Mef2 and cg drivers don't show over lapping expression is an important point. However it adds no new value to the storyline and is largely a well-executed control experiment. I feel it belongs in Supplemental figures. In Figure 3D-H, the color coding is difficult to keep track of amongst the figures and would recommend labeling the x-axis of these figures. This is also an issue in Figure S3B/E/G, where adding the genotype label instead of a color code would make the figures easier to follow.

We appreciate the suggestion. Labeling in Figure 3D-E-G-H were changed. The genotypes were labeled in the heat maps of Figure 5—figure supplement 1 too.

4. Figure 4, and Figure 2 convey the same message, Dp function in muscles has a systemic effect on Fat body. Hence it seems they belong in the same figure.

Since the genetic approaches used in these two figures are very different it may lead to some confusions for the general audience. That is why we kept them in separate figures. I hope the reviewer would agree.

5. Scale bars in figures are required.

Scale was added to the other panels of Figure 4A, Figure 6F, and Figure 5—figure supplement 1E

Text suggestions:– Lines 102-107 is a run-on sentence that should be broken down.

We appreciate the suggestion; sentence is now broken-down.

– Line 174 should reference the figure which is being described.

The reference to the panel was added.

– Paper is very jargon-heavy and requires a definition for a broader audience. One abbreviation that isn't defined in the paper is APF.

The abbreviation APF is now removed from the manuscript.